# On Representing Mixed-Integer Linear Programs by Graph Neural Networks

**Ziang Chen** [*]
Department of Mathematics, Duke University
Durham, NC 27708
ziang@math.duke.edu

**Jialin Liu** [†]
Damo Academy, Alibaba US
Bellevue, WA 98004
jialin.liu@alibaba-inc.com

**Xinshang Wang**
Damo Academy, Alibaba US
Antai College of Economics and Management, Shanghai Jiao Tong University
Bellevue, WA 98004
xinshang.w@alibaba-inc.com

**Jianfeng Lu**
Departments of Mathematics, Physics,
and Chemistry, Duke University
Durham, NC 27708
jianfeng@math.duke.edu

**Wotao Yin**
Damo Academy, Alibaba US
Bellevue, WA 98004
wotao.yin@alibaba-inc.com

## Abstract

While Mixed-integer linear programming (MILP) is NP-hard in general, practical MILP has received roughly 100–fold speedup in the past twenty years. Still, many classes of MILPs quickly become unsolvable as their sizes increase, motivating researchers to seek new acceleration techniques for MILPs. With deep learning, they have obtained strong empirical results, and many results were obtained by applying graph neural networks (GNNs) to making decisions in various stages of MILP solution processes. This work discovers a fundamental limitation: there exist feasible and infeasible MILPs that all GNNs will, however, treat equally, indicating GNN's lacking power to express general MILPs. Then, we show that, by restricting the MILPs to unfoldable ones or by adding random features, there exist GNNs that can reliably predict MILP feasibility, optimal objective values, and optimal solutions up to prescribed precision. We conducted small-scale numerical experiments to validate our theoretical findings.

## 1 Introduction

Mixed-integer linear programming (MILP) is a type of optimization problems that minimize a linear objective function subject to linear constraints, where some or all variables must take integer values. MILP has a wide type of applications, such as transportation (Schouwenaars et al., 2001), control (Richards & How, 2005), scheduling (Floudas & Lin, 2005), etc. Branch and Bound (B&B) (Land & Doig, 1960), an algorithm widely adopted in modern solvers that exactly solves general MILPs to global optimality, unfortunately, has an exponential time complexity in the worst-case sense. To make MILP more practical, researchers have to analyze the features of each instance of interest based on their domain knowledge, and use such features to adaptively warm-start B&B or design the heuristics in B&B.

To automate such laborious process, researchers turn attention to Machine learning (ML) techniques in recent years (Bengio et al., 2021). The literature has reported some encouraging findings that a proper chosen ML model is able to learn some useful knowledge of MILP from data and generalize well to some similar but unseen instances. For example, one can learn fast approximations of Strong

---

[*]A major part of the work of Z. Chen was completed during his internship at Alibaba US DAMO Academy.
[†]Corresponding author.

Branching, an effective but time-consuming branching strategy usually used in B&B (Alvarez et al., 2014; Khalil et al., 2016; Zarpellon et al., 2021; Lin et al., 2022). One may also learn cutting strategies (Tang et al., 2020; Berthold et al., 2022; Huang et al., 2022), node selection/pruning strategies (He et al., 2014; Yilmaz & Yorke-Smith, 2020), or decomposition strategies (Song et al., 2020) with ML models. The role of ML models in those approaches can be summarized as: *approximating useful mappings or parameterizing key strategies* in MILP solvers, and these mappings/strategies usually take an MILP instance as input and output its key peroperties.

The graph neural network (GNN), due to its nice properties, say *permutation invariance*, is considered a suitable model to represent such mappings/strategies for MILP. More specifically, permutations on variables or constraints of an MILP do not essentially change the problem itself, reliable ML models such as GNNs should satisfy such properties, otherwise the model may overfit to the variable/constraint orders in the training data. Gasse et al. (2019) proposed that an MILP can be encoded into a bipartite graph on which one can use a GNN to approximate Strong Branching. Ding et al. (2020) proposed to represent MILP with a tripartite graph. Since then, GNNs have been adopted to represent mappings/strategies for MILP, for example, approximating Strong Branching (Gupta et al., 2020; Nair et al., 2020; Shen et al., 2021; Gupta et al., 2022), approximating optimal solution (Nair et al., 2020; Khalil et al., 2022), parameterizing cutting strategies (Paulus et al., 2022), and parameterizing branching strategies (Qu et al., 2022; Scavuzzo et al., 2022).

However, theoretical foundations in this direction still remain unclear. A key problem is the ability of GNN to approximate important mappings related with MILP. We ask the following questions:

| | |
|---|---|
| Is GNN able to predict whether an MILP is feasible? | (Q1) |
| Is GNN able to approximate the optimal objective value of an MILP? | (Q2) |
| Is GNN able to approximate an optimal solution of an MILP? | (Q3) |

To answer questions (Q1 - Q3), one needs theories of separation power and representation power of GNN. The ***separation power*** of GNN is measured by whether it can distinguish two non-isomorphic graphs. Given two graphs $G_1, G_2$, we say a mapping $F$ (e.g. a GNN) has strong separation power if $F(G_1) \neq F(G_2)$ as long as $G_1, G_2$ that are not the isomorphic. In our settings, since MILPs are represented by graphs, the separation power actually indicates the ability of GNN to distinguish two different MILP instances. The ***representation power*** of GNN refers to how well it can approximate mappings with permutation invariant properties. In our settings, we study whether GNN can map an MILP to its feasiblity, optimal objective value and an optimal solution. The separation power and representation power of GNNs are closely related to the Weisfeiler-Lehman (WL) test (Weisfeiler & Leman, 1968), a classical algorithm to identify whether two graphs are isomorphic or not. It has been shown that GNN has the same separation power with the WL test (Xu et al., 2019), and, based on this result, GNNs can universally approximate continuous graph-input mappings with separation power no stronger than the WL test (Azizian & Lelarge, 2021; Geerts & Reutter, 2022).

**Our contributions**    With the above tools in hand, one still cannot directly answer questions (Q1 - Q3) since the relationship between characteristics of general MILPs and properties of graphs are not clear yet. Although there are some works studying the representation power of GNN on some graph-related optimization problems (Sato et al., 2019; Loukas, 2020) and linear programming (Chen et al., 2022), representing general MILPs with GNNs are still not theoretically studied, to the best of our knowledge. Our contributions are listed below:

- (Limitation of GNNs for MILP) We show with an example that GNNs do not have strong enough separation power to distinguish any two different MILP instances. There exist two MILPs such that one of them is feasible while the other one is not, but, unfortunately, all GNNs treat them equally without detecting the essential difference between them. In fact, there are infinitely many pairs of MILP instances that can puzzle GNN.

- (Foldable and unfoldable MILP) We provide a precise mathematical description on what type of MILPs makes GNNs fail. These hard MILP instances are named as *foldable MILPs*. We prove that, for *unfoldable MILPs*, GNN has strong enough separation power and representation power to approximate the feasibility, optimal objective value and an optimal solution.

- (MILP with random features) To handle those foldable MILPs, we propose to append random features to the MILP-induced graphs. We prove that, with the random feature technique, the answers to questions (Q1 - Q3) are affirmative.

The answers to (Q1 - Q3) serve as foundations of a more practical question: whether GNNs are able to predict branching strategies or primal heuristics for MILP? Although the answers to (Q1) and (Q2) do not directly answer that practical question, they illustrate the possibility that GNNs can capture some key information of a MILP instance and have the capacity to suggest an adaptive branching strategy for each instance. To obtain a GNN-based strategy, practitioners should consider more factors: feature spaces, action spaces, training methods, generalization performance, etc., and some recent empirical studies show encouraging results on learning such a strategy (Gasse et al., 2019; Gupta et al., 2020; 2022; Nair et al., 2020; Shen et al., 2021; Qu et al., 2022; Scavuzzo et al., 2022; Khalil et al., 2022). The answer to (Q3) directly shows the possibility of learning primal heuristics. With proper model design and training methods, one could obtain competitive GNN-based primal heuristics (Nair et al., 2020; Ding et al., 2020).

The rest of this paper is organized as follows. We state some preliminaries in Section 2. The limitation of GNN for MILP is presented in Section 3 and we provide the descriptions of foldable and unfoldable MILPs in Section 4. The random feature technique is introduced in Section 5. Section 6 contains some numerical results and the whole paper is concluded in Section 7.

## 2 PRELIMINARIES

In this section, we introduce some preliminaries that will be used throughout this paper. Our notations, definitions, and examples follow Chen et al. (2022). Consider a general MILP problem defined with:

$$\min_{x \in \mathbb{R}^n} \ c^\top x, \quad \text{s.t.} \ Ax \circ b, \ l \leq x \leq u, \ x_j \in \mathbb{Z}, \ \forall \, j \in I, \tag{2.1}$$

where $A \in \mathbb{R}^{m \times n}$, $c \in \mathbb{R}^n$, $b \in \mathbb{R}^m$, $l \in (\mathbb{R} \cup \{-\infty\})^n$, $u \in (\mathbb{R} \cup \{+\infty\})^n$, and $\circ \in \{\leq, =, \geq\}^m$. The index set $I \subset \{1, 2, \ldots, n\}$ includes those indices $j$ where $x_j$ are constrained to be an integer. The feasible set is defined with $X_{\text{feasible}} := \{x \in \mathbb{R}^n | Ax \circ b, \ l \leq x \leq u, \ x_j \in \mathbb{Z}, \forall \, j \in I\}$, and we say an MILP is *infeasible* if $X_{\text{feasible}} = \emptyset$ and *feasible* otherwise. For feasible MILPs, $\inf\{c^\top x : x \in X_{\text{feasible}}\}$ is named as the *optimal objective value*. If there exists $x^* \in X_{\text{feasible}}$ with $c^\top x^* \leq c^\top x, \ \forall \, x \in X_{\text{feasible}}$, then we say that $x^*$ is an *optimal solution*. It is possible that the objective value is arbitrarily good, i.e., for any $R > 0$, $c^\top \hat{x} < -R$ holds for some $\hat{x} \in X_{\text{feasible}}$. In this case, we say the MILP is *unbounded* or its optimal objective value is $-\infty$.

### 2.1 MILP AS A WEIGHTED BIPARTITE GRAPH WITH VERTEX FEATURES

Inspired by Gasse et al. (2019) and following Chen et al. (2022), we represent MILP by *weighted bipartite graphs*. The vertex set of such a graph is $V \cup W$, where $V = \{v_1, v_2, \ldots, v_m\}$ with $v_i$ representing the $i$-th constraint, and $W = \{w_1, w_2, \ldots, w_m\}$ with $w_j$ representing the $j$-th variable. To fully represent all information in (2.1), we associate each vertex with features: The vertex $v_i \in V$ is equipped with a feature vector $h_i^V = (b_i, \circ_i)$ that is chosen from $\mathcal{H}^V = \mathbb{R} \times \{\leq, =, \geq\}$. The vertex $w_j \in W$ is equipped with a feature vector $h_j^W = (c_j, l_j, u_j, \tau_j)$, where $\tau_j = 1$ if $j \in I$ and $\tau_j = 0$ otherwise. The feature $h_j^W$ is in the space $\mathcal{H}^W = \mathbb{R} \times (\mathbb{R} \cup \{-\infty\}) \times (\mathbb{R} \cup \{+\infty\}) \times \{0, 1\}$. The edge $E_{i,j} \in \mathbb{R}$ connects $v_i \in V$ and $w_j \in W$, and its value is defined with $E_{i,j} = A_{i,j}$. Note that there is no edge connecting vertices in the same vertex group ($V$ or $W$) and $E_{i,j} = 0$ if there is no connection between $v_i$ and $w_j$. Thus, the whole graph is denoted as $G = (V \cup W, E)$, and we denote $\mathcal{G}_{m,n}$ as the collection of all such weighted bipartite graphs whose two vertex groups have size $m$ and $n$, respectively. Finally, we define $\mathcal{H}_m^V := (\mathcal{H}^V)^m$ and $\mathcal{H}_n^W := (\mathcal{H}^W)^n$, and stack all the vertex features together as $H = (h_1^V, h_2^V, \ldots, h_m^V, h_1^W, h_2^W, \ldots, h_n^W) \in \mathcal{H}_m^V \times \mathcal{H}_n^W$. Then a weighted bipartite graph with vertex features $(G, H) \in \mathcal{G}_{m,n} \times \mathcal{H}_m^V \times \mathcal{H}_n^W$ contains all information in the MILP problem (2.1), and we name such a graph as an *MILP-induced graph* or *MILP-graph*. [1] If $I = \emptyset$, the feature $\tau_j$ can be dropped and the graphs reduce to LP-graphs in Chen et al. (2022). We provide an example of MILP-graph in Figure 1.

### 2.2 GRAPH NEURAL NETWORKS WITH MESSAGE PASSING FOR MILP-GRAPHS

To represent properties of the whole graph, one needs to build a GNN that maps $(G, H)$ to a real number: $\mathcal{G}_{m,n} \times \mathcal{H}_m^V \times \mathcal{H}_n^W \to \mathbb{R}$; to represent properties of each vertex in $W$ (or represent properties for each variable), one needs to build a GNN that maps $(G, H)$ to a vector: $\mathcal{G}_{m,n} \times \mathcal{H}_m^V \times \mathcal{H}_n^W \to \mathbb{R}^n$. The GNNs used in this paper can be constructed with the following three steps:

---

[1] In Gasse et al. (2019) and other related empirical studies, the feature spaces $\mathcal{H}^V$, $\mathcal{H}^W$ are more complicated than those defined in this paper. We only keep the most basic features here for simplicity of analysis.

$$\min_{x\in\mathbb{R}^2} \; x_1 + x_2,$$

$$\text{s.t. } x_1 + 3x_2 \geq 1, \; x_1 + x_2 \geq 1,$$
$$x_1 \leq 3, \; x_2 \leq 5, \; x_2 \in \mathbb{Z}.$$

$h_1^V = (1, \geq)$  $v_1$ —1— $w_1$  $h_1^W = (1, -\infty, 3, 0)$

$h_2^V = (1, \geq)$  $v_2$ —1— $w_2$  $h_2^W = (1, -\infty, 5, 1)$

Figure 1: An example of MILP-graph

(1) Initial mapping at level $l = 0$. Let $p^0$ and $q^0$ be learnable embedding mappings, then the embedded features are $s_i^0 = p^0(h_i^V)$ and $t_j^0 = q^0(h_j^W)$ for $i = 1, 2, \ldots, m$ and $j = 1, 2, \ldots, n$.

(2) Message-passing layers for $l = 1, 2, \ldots, L$. Given learnable mappings $f^l, g^l, p^l, q^l$, one can update vertex features by the following formulas for all $i = 1, 2, \ldots, m$ and $j = 1, 2, \ldots, n$:

$$s_i^l = p^l\left(s_i^{l-1}, \sum_{j=1}^{n} E_{i,j} f^l(t_j^{l-1})\right), \quad t_j^l = q^l\left(t_j^{l-1}, \sum_{i=1}^{m} E_{i,j} g^l(s_i^{l-1})\right).$$

(3a) Last layer for graph-level output. Define a learnable mapping $r_G$ and the output is a scalar $y_G = r_G(\bar{s}^L, \bar{t}^L)$, where $\bar{s}^L = \sum_{i=1}^{m} s_i^L$ and $\bar{t}^L = \sum_{j=1}^{n} t_j^L$.

(3b) Last layer for node-level output. Define a learnable mapping $r_W$ and the output is a vector $y \in \mathbb{R}^n$ with entries being $y_j = r_W(\bar{s}^L, \bar{t}^L, t_j^L)$ for $j = 1, 2, \ldots, n$.

Throughout this paper, we require all learnable mappings to be continuous following the same settings as in Chen et al. (2022); Azizian & Lelarge (2021). In practice, one may parameterize those mappings with multi-layer perceptions (MLPs). Define $\mathcal{F}_{\text{GNN}}$ as the set of all GNN mappings connecting layers (1), (2), and (3a). $\mathcal{F}_{\text{GNN}}^W$ is defined similarly by replacing (3a) with (3b).

## 2.3 MAPPINGS TO REPRESENT MILP CHARACTERISTICS

Now we introduce the mappings that are what we aim to approximate by GNNs. With the definitions, we will revisit questions (Q1-Q3) and describe them in a mathematically precise way.

**Feasibility mapping** We first define the following mapping that indicates the feasibility of MILP:

$$\Phi_{\text{feas}} : \mathcal{G}_{m,n} \times \mathcal{H}_m^V \times \mathcal{H}_n^W \rightarrow \{0, 1\},$$

where $\Phi_{\text{feas}}(G, H) = 1$ if $(G, H)$ corresponds to a feasible MILP and $\Phi_{\text{feas}}(G, H) = 0$ otherwise.

**Optimal objective value mapping** We then define the following mapping that maps an MILP to its optimal objective value:

$$\Phi_{\text{obj}} : \mathcal{G}_{m,n} \times \mathcal{H}_m^V \times \mathcal{H}_n^W \rightarrow \mathbb{R} \cup \{\infty, -\infty\},$$

where $\Phi_{\text{obj}}(G, H) = \infty$ implies infeasibility and $\Phi_{\text{obj}}(G, H) = -\infty$ implies unboundedness. Note that the optimal objective value for MILP may be an infimum that can never be achieved. An example would be $\min_{x\in\mathbb{Z}^2} x_1 + \pi x_2$, s.t. $x_1 + \pi x_2 \geq \sqrt{2}$. The optimal objective value is $\sqrt{2}$, since for any $\epsilon > 0$, there exists a feasible $x$ with $x_1 + \pi x_2 < \sqrt{2} + \epsilon$. However, there is no $x \in \mathbb{Z}^2$ such that $x_1 + \pi x_2 = \sqrt{2}$. Thus, the preimage $\Phi_{\text{obj}}^{-1}(\mathbb{R})$ cannot precisely describe all MILP instances with an optimal solution, it describes MILP problems with a finite optimal objective.

**Optimal solution mapping** To give a well-defined optimal solution mapping is much more complicated [2] since the optimal objective value, as we discussed before, may never be achieved in some cases. To handle this issue, we only consider the case that any component in $l$ or $u$ must be finite, which implies that an optimal solution exists as long as the MILP problem is feasible. More specifically, the vertex feature space is limited to $\widetilde{\mathcal{H}}_n^W = (\mathbb{R} \times \mathbb{R} \times \mathbb{R} \times \{0, 1\})^n \subset \mathcal{H}_n^W$ and we consider MILP problems taken from $\mathcal{D}_{\text{solu}} = (\mathcal{G}_{m,n} \times \mathcal{H}_m^V \times \widetilde{\mathcal{H}}_n^W) \cap \Phi_{\text{feas}}^{-1}(1)$. Note that $\Phi_{\text{feas}}^{-1}(1)$ describes the set of all feasible MILPs. Consequently, any MILP instance in $\mathcal{D}_{\text{solu}}$ admits at least one optimal solution. We can further define the following mapping which maps an MILP to exactly one of its optimal solutions:

$$\Phi_{\text{solu}} : \mathcal{D}_{\text{solu}} \backslash \mathcal{D}_{\text{foldable}} \rightarrow \mathbb{R}^n,$$

where $\mathcal{D}_{\text{foldable}}$ is a subset of $\mathcal{G}_{m,n} \times \mathcal{H}_m^V \times \mathcal{H}_n^W$ that will be introduced in Section 4. The full definition of $\Phi_{\text{solu}}$ is placed in Appendix C due to its tediousness.

---

[2] If we remove the integer constraints $I = \emptyset$ and let MILP reduces to linear programming (LP), the solution mapping will be easier to define. In this case, as long as the optimal objective value is finite, there must exist an optimal solution, and the optimal solution with the smallest $\ell_2$-norm is unique (Chen et al., 2022). Therefore, a mapping $\Phi_{\text{solu}}$, which maps an LP to its optimal solution with the smallest $\ell_2$-norm, is well defined on $\Phi_{\text{obj}}^{-1}(\mathbb{R})$.

**Invariance and equivariance** Now we discuss some properties of the three defined mappings. Mappings $\Phi_{\text{feas}}$ and $\Phi_{\text{obj}}$ are *permutation invariant* because the feasibility and optimal objective of an MILP would not change if the variables or constraints are reordered. We say the mapping $\Phi_{\text{solu}}$ is *permutation equivariant* because the solution of an MILP should be reordered consistently with the permutation on the variables. Now we define $S_m$ as the group contains all permutations on the constraints of MILP and $S_n$ as the group contains all permutations on the variables. For any $\sigma_V \in S_m$ and $\sigma_W \in S_n$, $(\sigma_V, \sigma_W) * (G, H)$ denotes the reordered MILP-graph with permutations $\sigma_V, \sigma_W$. It is clear that both $\Phi_{\text{feas}}$ and $\Phi_{\text{obj}}$ are permutation invariant in the following sense:

$$\Phi_{\text{feas}}((\sigma_V, \sigma_W) * (G, H)) = \Phi_{\text{feas}}(G, H), \quad \Phi_{\text{obj}}((\sigma_V, \sigma_W) * (G, H)) = \Phi_{\text{obj}}(G, H),$$

for all $\sigma_V \in S_m$, $\sigma_W \in S_n$, and $(G, H) \in \mathcal{G}_{m,n} \times \mathcal{H}_m^V \times \mathcal{H}_n^W$. In addition, $\Phi_{\text{solu}}$ is permutation equivariant: $\Phi_{\text{solu}}((\sigma_V, \sigma_W) * (G, H)) = \sigma_W(\Phi_{\text{solu}}(G, H))$, for all $\sigma_V \in S_m$, $\sigma_W \in S_n$, and $(G, H) \in \mathcal{D}_{\text{solu}} \backslash \mathcal{D}_{\text{foldable}}$. This will be discussed in Section C. Furthermore, one may check that any $F \in \mathcal{F}_{\text{GNN}}$ is invariant and any $F_W \in \mathcal{F}_{\text{GNN}}^W$ is equivariant.

**Revisiting questions** (Q1)**,** (Q2) **and** (Q3)   With the definitions above, questions (Q1),(Q2) and (Q3) actually ask: Given any finite set $\mathcal{D} \subset \mathcal{G}_{m,n} \times \mathcal{H}_m^V \times \mathcal{H}_n^W$, is there $F \in \mathcal{F}_{\text{GNN}}$ such that $F$ well approximates $\Phi_{\text{feas}}$ or $\Phi_{\text{obj}}$ on set $\mathcal{D}$? Given any finite set $\mathcal{D} \subset \mathcal{D}_{\text{solu}} \backslash \mathcal{D}_{\text{foldable}}$, is there $F_W \in \mathcal{F}_{\text{GNN}}^W$ such that $F_W$ is close to $\Phi_{\text{solu}}$ on set $\mathcal{D}$?

## 3   DIRECTLY APPLYING GNNs MAY FAIL ON GENERAL DATASETS

In this section, we show a limitation of GNN to represent MILP. To well approximate the mapping $\Phi_{\text{feas}}$, $\mathcal{F}_{\text{GNN}}$ should have *stronger separation power* than $\Phi_{\text{feas}}$: for any two MILP instances $(G, H), (\hat{G}, \hat{H}) \in \mathcal{G}_{m,n} \times \mathcal{H}_m^V \times \mathcal{H}_n^W$,

$$\Phi_{\text{feas}}(G, H) \neq \Phi_{\text{feas}}(\hat{G}, \hat{H}) \text{ implies } F(G, H) \neq F(\hat{G}, \hat{H}) \text{ for some } F \in \mathcal{F}_{\text{GNN}}.$$

In another word, as long as two MILP instances have different feasibility, there should be some GNNs that can detect that and give different outputs. Otherwise, we way that the whole GNN family $\mathcal{F}_{\text{GNN}}$ *cannot distinguish* two MILPs with different feasibility, hence, GNN cannot well approximate $\Phi_{\text{feas}}$. This motivate us to study the separation power of GNN for MILP.

In the literature, the separation power of GNN is usually measured by so-called Weisfeiler-Lehman (WL) test (Weisfeiler & Leman, 1968). We present a variant of WL test specially modified for MILP in Algorithm 1 that follows the same lines as in Chen et al. (2022, Algorithm 1), where each vertex is labeled with a color. For example, $v_i \in V$ is initially labeled with $C_i^{0,V}$ based on its feature $h_i^V$ by hash function $\text{HASH}_{0,V}$ that is assumed to be powerful enough such that it labels the vertices that have distinct information with distinct colors. After that, each vertex iteratively updates its color, based on its own color and information from its neighbors. Roughly speaking, *as long as two vertices in a graph are essentially different, they will get distinct colors finally.* The output of Algorithm 1 contains all vertex colors $\{\{C_i^{L,V}\}\}_{i=0}^m, \{\{C_j^{L,W}\}\}_{j=0}^n$, where $\{\{\}\}$ refers to a multiset in which the multiplicity of an element can be greater than one. Such approach is also named as color refinement (Berkholz et al., 2017; Arvind et al., 2015; 2017).

---

**Algorithm 1** WL test for MILP-Graphs[3](denoted by $\text{WL}_{\text{MILP}}$)

---

**Require:** A graph instance $(G, H) \in \mathcal{G}_{m,n} \times \mathcal{H}_m^V \times \mathcal{H}_n^W$ and iteration limit $L > 0$.
1: Initialize with $C_i^{0,V} = \text{HASH}_{0,V}(h_i^V)$, $C_j^{0,W} = \text{HASH}_{0,W}(h_j^W)$.
2: **for** $l = 1, 2, \cdots, L$ **do**
3:    $C_i^{l,V} = \text{HASH}_{l,V}\left(C_i^{l-1,V}, \sum_{j=1}^n E_{i,j}\text{HASH}'_{l,W}\left(C_j^{l-1,W}\right)\right)$.
4:    $C_j^{l,W} = \text{HASH}_{l,W}\left(C_j^{l-1,W}, \sum_{i=1}^m E_{i,j}\text{HASH}'_{l,V}\left(C_i^{l-1,V}\right)\right)$.
5: **end for**
6: **return** The multisets containing all colors $\{\{C_i^{L,V}\}\}_{i=0}^m, \{\{C_j^{L,W}\}\}_{j=0}^n$.

---

[3]In Algorithm 1, the hash functions $\{\text{HASH}_{l,V}, \text{HASH}_{l,W}\}_{l=0}^L$ can be any injective mappings defined on given domains, their output spaces consist of all possible vertex colors. The other hash functions $\{\text{HASH}'_{l,V}, \text{HASH}'_{l,W}\}_{l=0}^L$ are required to injectively map vertex colors to a linear space because we need to define sum and scalar multiplication on their outputs. Actually, Lemma 3.2 also applies on the WL test with a more general update scheme: $C_i^{l,V} = \text{HASH}_{l,V}(C_i^{l-1,V}, \{\{\text{HASH}'_{l,W}(C_j^{l-1,W}, E_{i,j})\}\}), C_j^{l,W} = \text{HASH}_{l,W}(C_j^{l-1,W}, \{\{\text{HASH}'_{l,V}(C_i^{l-1,V}, E_{i,j})\}\})$.

Unfortunately, there exist some non-isomorphic graph pairs that WL test fail to distinguish (Douglas, 2011). Throughout this paper, we use $(G, H) \sim (\hat{G}, \hat{H})$ to denote that $(G, H)$ and $(\hat{G}, \hat{H})$ *cannot be distinguished by the WL test*, i.e., $\mathrm{WL}_{\mathrm{MILP}}((G, H), L) = \mathrm{WL}_{\mathrm{MILP}}((\hat{G}, \hat{H}), L)$ holds for any $L \in \mathbb{N}$ and any hash functions. The following theorem indicates that $\mathcal{F}_{\mathrm{GNN}}$ actually has the same separation power with the WL test.

**Theorem 3.1** (Theorem 4.2 in Chen et al. (2022)). *For any $(G, H), (\hat{G}, \hat{H}) \in \mathcal{G}_{m,n} \times \mathcal{H}_m^V \times \mathcal{H}_n^W$, it holds that $(G, H) \sim (\hat{G}, \hat{H})$ if and only if $F(G, H) = F(\hat{G}, \hat{H}), \forall F \in \mathcal{F}_{GNN}$.*

Theorem 3.1 is stated and proved in Chen et al. (2022) for LP-graphs, but it actually also applies for MILP-graphs. The intuition for Theorem 3.1 is straightforward since the color refinement in WL test and the message-passing layer in GNNs have similar structure. The proof of Chen et al. (2022, Theorem 4.2) basically uses the fact that given finitely many inputs, there always exist hash functions or continuous functions that are injective and can generate linearly independent outputs, i.e., without collision. We also remark that the equivalence between the separation powers of GNNs and WL test has been investigated in some earlier literature (Xu et al., 2019; Azizian & Lelarge, 2021; Geerts & Reutter, 2022). Unfortunately, the following lemma reveals that the separation power of WL test is weaker than $\Phi_{\mathrm{feas}}$, and, consequently, GNN has weaker separation power than $\Phi_{\mathrm{feas}}$, on some specific MILP datasets.

**Lemma 3.2.** *There exist two MILP problems $(G, H)$ and $(\hat{G}, \hat{H})$ with one being feasible and the other one being infeasible, such that $(G, H) \sim (\hat{G}, \hat{H})$.*

*Proof of Lemma 3.2.* Consider two MILP problems and their induced graphs:

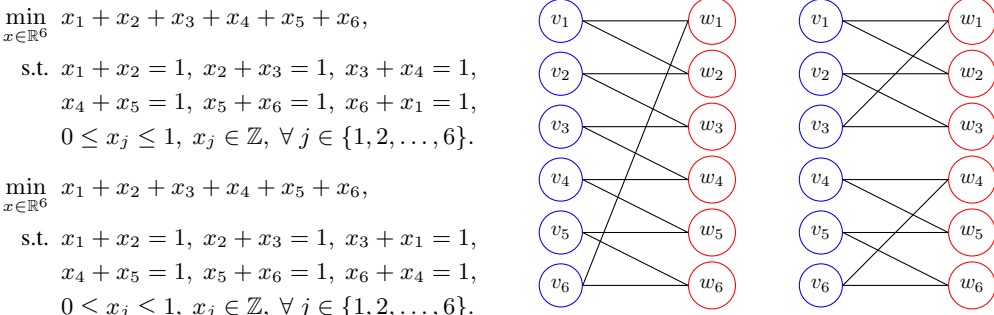

$$\min_{x \in \mathbb{R}^6} \ x_1 + x_2 + x_3 + x_4 + x_5 + x_6,$$
$$\text{s.t. } x_1 + x_2 = 1, \ x_2 + x_3 = 1, \ x_3 + x_4 = 1,$$
$$x_4 + x_5 = 1, \ x_5 + x_6 = 1, \ x_6 + x_1 = 1,$$
$$0 \le x_j \le 1, \ x_j \in \mathbb{Z}, \ \forall \, j \in \{1, 2, \ldots, 6\}.$$

$$\min_{x \in \mathbb{R}^6} \ x_1 + x_2 + x_3 + x_4 + x_5 + x_6,$$
$$\text{s.t. } x_1 + x_2 = 1, \ x_2 + x_3 = 1, \ x_3 + x_1 = 1,$$
$$x_4 + x_5 = 1, \ x_5 + x_6 = 1, \ x_6 + x_4 = 1,$$
$$0 \le x_j \le 1, \ x_j \in \mathbb{Z}, \ \forall \, j \in \{1, 2, \ldots, 6\}.$$

Figure 2: Two non-isomorphic MILP graphs that cannot be distinguished by WL test

The two MILP-graphs in Fugure 2 can not be distinguished by WL test, which can be proved by induction. First we consider the initial step in Algorithm 1. Based on the definitions in Section 2.2, we can explicitly write down the vertex features for each vertex here: $h_i^V = (1, =), 1 \le i \le 6$ and $h_j^W = (1, 0, 1, 1), 1 \le j \le 6$. Since all the vertices in $V$ share the same information, they are labeled with an uniform color $C_1^{0,V} = C_2^{0,V} \cdots = C_6^{0,V}$ (We use blue in Figure 2), whatever the hash functions we choose. With the same argument, one would obtain $C_1^{0,W} = C_2^{0,W} \cdots = C_6^{0,W}$ and label all vertices in $W$ with red. Both of the two graphs will be initialized in such an approach. Assuming $\{C_i^{l,V}\}_{i=1}^6$ are all blue and $\{C_j^{l,W}\}_{j=1}^6$ are all red, one will obtain *for both of the graphs in Figure 2* that $C_1^{l+1,V} = C_2^{l+1,V} \cdots = C_6^{l+1,V}$ and $C_1^{l+1,W} = C_2^{l+1,W} \cdots = C_6^{l+1,W}$ based on the update rule in Algorithm 1, because each blue vertex has two red neighbors and each red vertex has two blue neighbors, and each edge connecting a blue vertex with a red one has weight $1$. This concludes that, one cannot distinguish the two graphs by checking the outputs of the WL test.

However, the first MILP problem is feasible, since $x = (0, 1, 0, 1, 0, 1)$ is a feasible solution, while the second MILP problem is infeasible, since the constraints imply $3 = 2(x_1 + x_2 + x_3) \in 2\mathbb{Z}$, which is a contradiction. $\square$

For those instances in Figure 2, GNNs may struggle to predict a meaningful neural diving or branching strategy. In neural diving (Nair et al., 2020), one usually trains a GNN to predict an estimated solution; in neural branching (Gasse et al., 2019), one usually trains a GNN to predict a ranking score for each variable and uses that score to decide which variable is selected to branch first. However, the analysis above illustrates that the WL test or GNNs cannot distinguish the 6 variables, and one cannot get a meaningful ranking score or estimated solution.

## 4   UNFOLDABLE MILP PROBLEMS

We prove with example in Section 3 that one may not expect good performance of GNN to approximate $\Phi_{\text{feas}}$ on a general dataset of MILP problems. It's worth to ask: is it possible to describe the common characters of those hard examples? If so, one may restrict the dataset by removing such instances, and establish a strong separation/representation power of GNN on that restricted dataset. The following definition provides a rigorous description of such MILP instances.

**Definition 4.1** (Foldable MILP). *Given any MILP instance, one would obtain vertex colors $\{C_i^{l,V}, C_j^{l,W}\}_{l,i,j}$ by running Algorithm 1. We say that an MILP instance can be folded, or is foldable, if there exist $1 \leq i, i' \leq m$ or $1 \leq j, j' \leq n$ such that*

$$C_i^{l,V} = C_{i'}^{l,V}, \ \ i \neq i', \quad \text{or} \quad C_j^{l,W} = C_{j'}^{l,W}, \ \ j \neq j',$$

*for any $l \in \mathbb{N}$ and any hash functions. In another word, at least one color in the multisets generated by the WL test always has a multiplicity greater than 1. Furthermore, we denote $\mathcal{D}_{foldable} \subset \mathcal{G}_{m,n} \times \mathcal{H}_m^V \times \mathcal{H}_n^W$ as the collection of all $(G, H) \in \mathcal{G}_{m,n} \times \mathcal{H}_m^V \times \mathcal{H}_n^W$ that can be folded.*

For example, the MILP in Figure 1 is not foldable, while the two MILP instances in Figure 2 are both foldable. The foldable examples in Figure 2 have been analyzed in the proof of Lemma 3.2, and now we provide some analysis of the example in Figure 1 here. Since the vertex features are distinct $h_1^W \neq h_2^W$, one would obtain $C_1^{0,W} \neq C_2^{0,W}$ as long as the hash function $\text{HASH}_{0,W}$ is injective. Although $h_1^V = h_2^V$ and hence $C_1^{0,V} = C_2^{0,V}$, the neighborhood information of $v_1$ and $v_2$ are different due to the difference of the edge weights. One could obtain $C_1^{1,V} \neq C_2^{1,V}$ by properly choosing $\text{HASH}_{0,W}, \text{HASH}'_{1,W}, \text{HASH}_{1,V}$, which concludes the unfoldability.

We prove that, as long as those foldable MILPs are removed, GNN is able to accurately predict the feasibility of all MILP instances in a dataset with finite samples. We use $(F(G, H) > 1/2)$ as the criteria, and the indicator $\mathbb{I}_{F(G,H)>1/2} = 1$ if $F(G, H) > 1/2$; $\mathbb{I}_{F(G,H)>1/2} = 0$ otherwise.

**Theorem 4.2.** *For any finite dataset $\mathcal{D} \subset \mathcal{G}_{m,n} \times \mathcal{H}_m^V \times \mathcal{H}_n^W \backslash \mathcal{D}_{foldable}$, there exists $F \in \mathcal{F}_{GNN}$ such that*

$$\mathbb{I}_{F(G,H)>1/2} = \Phi_{feas}(G, H), \quad \forall (G, H) \in \mathcal{D}.$$

Similar results also hold for the optimal objective value mapping $\Phi_{\text{obj}}$ and the optimal solution mapping $\Phi_{\text{solu}}$ and we list the results below. All the proofs of theorems are deferred to the appendix.

**Theorem 4.3.** *Let $\mathcal{D} \subset \mathcal{G}_{m,n} \times \mathcal{H}_m^V \times \mathcal{H}_n^W \backslash \mathcal{D}_{foldable}$ be a finite dataset. For any $\delta > 0$, there exists $F_1 \in \mathcal{F}_{GNN}$ such that*

$$\mathbb{I}_{F_1(G,H)>1/2} = \mathbb{I}_{\Phi_{obj}(G,H) \in \mathbb{R}}, \quad \forall (G, H) \in \mathcal{D},$$

*and there exists some $F_2 \in \mathcal{F}_{GNN}$ such that*

$$|F_2(G, H) - \Phi_{obj}(G, H)| < \delta, \quad \forall (G, H) \in \mathcal{D} \cap \Phi_{obj}^{-1}(\mathbb{R}).$$

**Theorem 4.4.** *Let $\mathcal{D} \subset \mathcal{D}_{solu} \backslash \mathcal{D}_{foldable} \subset \mathcal{G}_{m,n} \times \mathcal{H}_m^V \times \mathcal{H}_n^W \backslash \mathcal{D}_{foldable}$ be a finite dataset. For any $\delta > 0$, there exists $F_W \in \mathcal{F}_{GNN}^W$ such that*

$$\|F_W(G, H) - \Phi_{solu}(G, H)\| < \delta, \quad \forall (G, H) \in \mathcal{D}.$$

Actually, conclusions in Theorems 4.2, 4.3 and 4.4 can be strengthened. The dataset $\mathcal{D}$ in the three theorems can be replaced with a measurable set with finite measure, which may contains infinitely many instances. Those strengthened theorems and their proofs can be found in Appendix A.

Let us also mention that the foldability of an MILP instance may depend on the feature spaces $\mathcal{H}^V$ and $\mathcal{H}^W$. If more features are appended (see Gasse et al. (2019)), i.e., $h_i^V$ and $h_j^W$ have more entries, and then a foldable MILP problem may become unfoldable. In addition, all the analysis here works for any topological linear spaces $\mathcal{H}^V$ and $\mathcal{H}^W$, as long as $\Phi_{\text{feas}}, \Phi_{\text{obj}}$, and $\Phi_{\text{solu}}$ can be verified as measurable and permutation-invariant/equivariant mappings.

## 5   SYMMETRY BREAKING TECHNIQUES VIA RANDOM FEATURES

Although we prove that GNN is able to approximate $\Phi_{\text{feas}}, \Phi_{\text{obj}}, \Phi_{\text{solu}}$ to any given precision for those unfoldable MILP instances, practitioners cannot benefit from that if there are foldable MILPs in their set of interest. For example, for around $1/4$ of the problems in MIPLIB 2017 (Gleixner et al., 2021), the number of colors generated by WL test is smaller than one half of the number of vertices, respect to the $\mathcal{H}^V$ and $\mathcal{H}^W$ used in this paper. To resolve this issue, we introduce a technique inspired by

Abboud et al. (2020); Sato et al. (2021). More specifically, we append the vertex features with an additional random feature $\omega$ and define a type of random GNNs as follows.

Let $\Omega = [0,1]^m \times [0,1]^n$ and let $(\Omega, \mathcal{F}_\Omega, \mathbb{P})$ be the probability space corresponding to the uniform distribution $\mathcal{U}(\Omega)$. The class of random graph neural network $\mathcal{F}_{\text{GNN}}^R$ with scalar output is the collection of functions

$$
\begin{aligned}
F_R : \mathcal{G}_{m,n} \times \mathcal{H}_m^V \times \mathcal{H}_n^W \times \Omega &\to \quad\ \mathbb{R}, \\
(G, H, \omega) &\mapsto F_R(G, H, \omega),
\end{aligned}
$$

which is defined in the same way as $\mathcal{F}_{\text{GNN}}$ with input space being $\mathcal{G}_{m,n} \times \mathcal{H}_m^V \times \mathcal{H}_n^W \times \Omega \cong \mathcal{G}_{m,n} \times (\mathcal{H}^V \times [0,1])^m \times (\mathcal{H}^W \times [0,1])^n$ and $\omega$ being sampled from $\mathcal{U}(\Omega)$. The class of random graph neural network $\mathcal{F}_{\text{GNN}}^{W,R}$ with vector output is the collection of functions

$$
\begin{aligned}
F_{W,R} : \mathcal{G}_{m,n} \times \mathcal{H}_m^V \times \mathcal{H}_n^W \times \Omega &\to \quad\ \mathbb{R}^n, \\
(G, H, \omega) &\mapsto F_{W,R}(G, H, \omega),
\end{aligned}
$$

which is defined in the same way as $\mathcal{F}_{\text{GNN}}^W$ with input space being $\mathcal{G}_{m,n} \times \mathcal{H}_m^V \times \mathcal{H}_n^W \times \Omega \cong \mathcal{G}_{m,n} \times (\mathcal{H}^V \times [0,1])^m \times (\mathcal{H}^W \times [0,1])^n$ and $\omega$ being sampled from $\mathcal{U}(\Omega)$.

By the definition of graph neural networks, $F_R$ is permutation invariant and $F_{W,R}$ is permutation equivariant, where the permutations $\sigma_V$ and $\sigma_W$ act on $(\mathcal{H}^V \times [0,1])^m$ and $(\mathcal{H}^W \times [0,1])^n$. We also write $F_R(G, H) = F_R(G, H, \omega)$ and $F_{W,R}(G, H) = F_{W,R}(G, H, \omega)$ as random variables. The theorem below states that, by adding random features, GNNs have sufficient power to represent MILP feasibility, even including those foldable MILPs. The intuition is that by appending additional random features, with probability one, each vertex will have distinct features and the resulting MILP-graph is hence unfoldable, even if it is foldable originally.

**Theorem 5.1.** *Let $\mathcal{D} \subset \mathcal{G}_{m,n} \times \mathcal{H}_m^V \times \mathcal{H}_n^W$ be a finite dataset. For any $\epsilon > 0$, there exists $F_R \in \mathcal{F}_{GNN}^R$, such that*

$$
\mathbb{P}\left(\mathbb{I}_{F_R(G,H)>1/2} \neq \Phi_{feas}(G,H)\right) < \epsilon, \quad \forall\, (G,H) \in \mathcal{D}.
$$

Similar results also hold for the optimal objective value and the optimal solution.

**Theorem 5.2.** *Let $\mathcal{D} \subset \mathcal{G}_{m,n} \times \mathcal{H}_m^V \times \mathcal{H}_n^W$ be a finite dataset. For any $\epsilon, \delta > 0$, there exists $F_{R,1} \in \mathcal{F}_{GNN}^R$, such that*

$$
\mathbb{P}\left(\mathbb{I}_{F_{R,1}(G,H)>1/2} \neq \mathbb{I}_{\Phi_{obj}(G,H) \in \mathbb{R}}\right) < \epsilon, \quad \forall\, (G,H) \in \mathcal{D},
$$

*and there exists $F_{R,2} \in \mathcal{F}_{GNN}^R$, such that*

$$
\mathbb{P}\left(|F_{R,2}(G,H) - \Phi_{obj}(G,H)| > \delta\right) < \epsilon, \quad \forall\, (G,H) \in \mathcal{D} \cap \Phi_{obj}^{-1}(\mathbb{R}).
$$

**Theorem 5.3.** *Let $\mathcal{D} \subset \Phi_{obj}^{-1}(\mathbb{R}) \subset \mathcal{G}_{m,n} \times \mathcal{H}_m^V \times \mathcal{H}_n^W$ be a finite dataset. For any $\epsilon, \delta > 0$, there exists $F_{W,R} \in \mathcal{F}_{GNN}^{W,R}$, such that*

$$
\mathbb{P}\left(\|F_{W,R}(G,H) - \Phi_{solu}(G,H)\| > \delta\right) < \epsilon, \quad \forall\, (G,H) \in \mathcal{D}.
$$

The idea behind those theorems is that GNNs can distinguish $(G, H, \omega)$ and $(\hat{G}, \hat{H}, \hat{\omega})$ with high probability, even if they cannot distinguish $(G, H)$ and $(\hat{G}, \hat{H})$. How to choose the random feature in training is significant. In practice, one may generate one random feature vector $\omega \in [0,1]^m \times [0,1]^n$ for all MILP instances. This setting leads to efficiency in training GNN models, but the trained GNNs cannot be applied to datasets with MILP problems of different sizes. Another practice is sampling several independent random features for each MILP instance, but one may suffer from difficulty in training. Such a trade-off also occurs in other GNN tasks (Loukas, 2019; Balcilar et al., 2021). In this paper, we generate one random vector for all instances (both training and testing instances) to directly validate our theorems. How to balance the trade-off in practice will be an interesting future topic.

## 6 NUMERICAL EXPERIMENTS

In this section, we experimentally validate our theories on some small-scale examples with $m = 6$ and $n = 20$. We first randomly generate two datasets $\mathcal{D}_1$ and $\mathcal{D}_2$. Set $\mathcal{D}_1$ consists of 1000 randomly generate MILPs that are all unfoldable, and there are 460 feasible MILPs whose optimal solutions are attachable while the others are all infeasible. Set $\mathcal{D}_2$ consists of 1000 randomly generate MILPs that are all foldable and similar to the example provided in Figure 2, and there are 500 feasible

MILPs with attachable optimal solution while the others are infeasible. We call SCIP (Bestuzheva et al., 2021a;b), a state-of-the-art non-commercial MILP solver, to obtain the feasibility and optimal solution for each instance. In our GNNs, we set the number of message-passing layers as $L = 2$ and parameterize all the learnable functions $f_{\mathrm{in}}^V, f_{\mathrm{in}}^W, f_{\mathrm{out}}, f_{\mathrm{out}}^W, \{f_l^V, f_l^W, g_l^V, g_l^W\}_{l=0}^L$ as multilayer perceptrons (MLPs). Our codes are modified from Gasse et al. (2019) and released to `https://github.com/liujl11git/GNN-MILP.git`. All the results reported in this section are obtained on the training sets, not an separate testing set. Generalization tests and details of the numerical experiments can be found in the appendix.

**Feasibility**    We first test whether GNN can represent the feasibility of an MILP and report our results in Figure 3. The orange curve with tag "Foldable MILPs" presents the training result of GNN on set $\mathcal{D}_2$. It's clear that GNN fails to distinguish the feasible and infeasible MILP pairs that are foldable, *whatever the GNN size we take*. However, if we train GNNs on those unfoldable MILPs in set $\mathcal{D}_1$, it's clear that the rate of errors goes to zero, as long as the size of GNN is large enough (the number of GNN parameters is large enough). This result validates Theorem 4.2 and the first conclusion in Theorem 4.3: the existence of GNNs that can accurately predict whether an

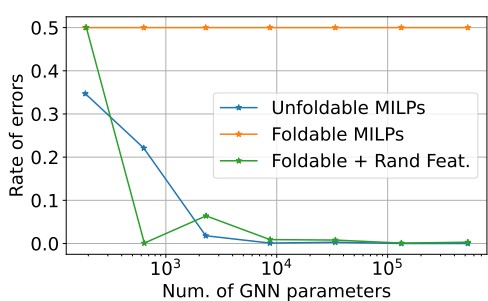

Figure 3: Feasibility

MILP is feasible (or whether an MILP has a finite optimal objective value). Finally, we append additional random features to the vertex features in GNN. As the green curve with tag "Foldable + Rand Feat." shown, GNN can perfectly fit the foldable data $\mathcal{D}_2$, which validates Theorem 5.1 and the first conclusion in Theorem 5.2.

**Optimal value and solution**    Then we take the feasible instances from sets $\mathcal{D}_1$ and $\mathcal{D}_2$ and form new datasets $\mathcal{D}_1^{\mathrm{feasible}}$ and $\mathcal{D}_2^{\mathrm{feasible}}$, respectively. On $\mathcal{D}_1^{\mathrm{feasible}}$ and $\mathcal{D}_2^{\mathrm{feasible}}$, we validate that GNN is able to approximate the optimal objective value and one optimal solution. Figure 4a shows that, by restricting datasets to unfoldable instances, or by appending random features to the graph, one can train a GNN that has arbitrarily small approximation error for the optimal objective value. Such conclusions validates Theorems 4.3 and 5.2. Figure 4b shows that GNNs can even approximate an optimal solution of MILP, though it requires a much larger size than the case of approximating optimal objective. Theorems 4.4 and 5.3 are validated.

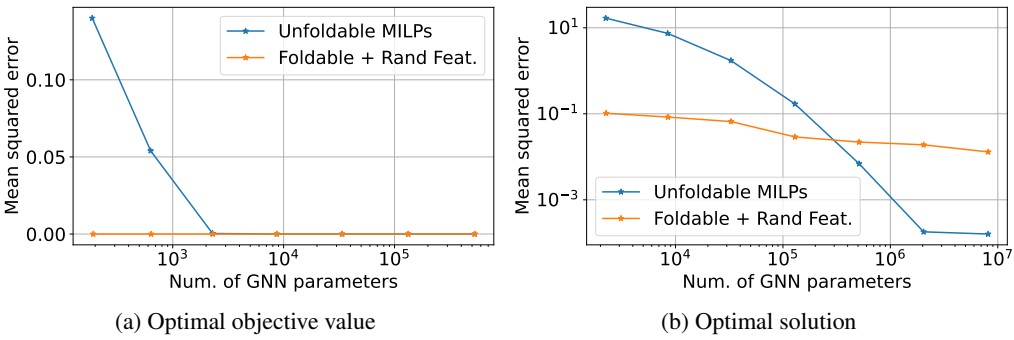

(a) Optimal objective value

(b) Optimal solution

Figure 4: GNN can approximate $\Phi_{\mathrm{obj}}$, and $\Phi_{\mathrm{solu}}$

## 7    CONCLUSION

This work investigates the expressive power of graph neural networks for representing mixed-integer linear programming problems. It is found that the separation power of GNNs bounded by that of WL test is not sufficient for foldable MILP problems, and in contrast, we show that GNNs can approximate characteristics of unfoldable MILP problems with arbitrarily small error. To get rid of the requirement on unfoldability which may not be true in practice, a technique of appending random feature is discuss with theoretical guarantee. We conduct numerical experiments for all the theory. This paper will contribute to the recently active field of applying GNNs for MILP solvers.

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

# A    PROOFS FOR SECTION 4

In this section, we prove the results in Section 4, i.e., graph neural networks can approximate $\Phi_{\text{feas}}$, $\Phi_{\text{obj}}$, and $\Phi_{\text{solu}}$, on finite datasets of unfoldable MILPs with arbitrarily small error. We would consider more general results on finite-measure subset of $\mathcal{G}_{m,n} \times \mathcal{H}_m^V \times \mathcal{H}_n^W$ which may involve infinite elements.

In our settings, foldability is actually a concept only depending on the MILP instance. In the main text, we utilize the tool of the WL test to define foldability here for readability. Here we present a more complex but equivalent definition merely with the language of MILP.

**Definition A.1** (Foldable MILP). *We say that the MILP problem (2.1) can be folded, or is foldable, if there exist $\mathcal{I} = \{I_1, I_2, \ldots, I_s\}$ and $\mathcal{J} = \{J_1, J_2, \ldots, J_t\}$ that are partitions of $\{1, 2, \ldots, m\}$ and $\{1, 2, \ldots, n\}$, respectively, such that at least one of $\mathcal{I}$ and $\mathcal{J}$ is nontrivial, i.e., $s < m$ or $t < n$, and that the followings hold for any $p \in \{1, 2, \ldots, s\}$ and $q \in \{1, 2, \ldots, t\}$: vertices in $\{v_i : i \in I_p\}$ share the same feature in $\mathcal{H}^V$, vertices in $\{w_j : j \in J_q\}$ share the same feature in $\mathcal{H}^W$, and all column sums (and row sums) of the submatrix $(A_{i,j})_{i \in I_p, j \in J_q}$ are the same.*

According to Chen et al. (2022, Appendix A), the color refinement procedure of the WL test converges to the coarsest partitions $\mathcal{I}$ and $\mathcal{J}$ satisfying the conditions in Definition A.1, assuming no collision happens. Therefore, one can see that Definition 4.1 and Definition A.1 are equivalent.

Before processing to establish the approximation theorems for GNNs on $\mathcal{G}_{m,n} \times \mathcal{H}_m^V \times \mathcal{H}_n^W \backslash \mathcal{D}_{\text{foldable}}$, let us define the topology and the measure on $\mathcal{G}_{m,n} \times \mathcal{H}_m^V \times \mathcal{H}_n^W$.

**Topology and measure**    The space $\mathcal{G}_{m,n} \times \mathcal{H}_m^V \times \mathcal{H}_n^W$ can be equipped with some natural topology and measure. In the construction of $\mathcal{G}_{m,n} \times \mathcal{H}_m^V \times \mathcal{H}_n^W$, let all Euclidean spaces be equipped with standard topology and the Lebesgue measure; let all discrete spaces be equipped with the discrete topology and the discrete measure with each elements being of measure 1. Then let all unions be disjoint unions and let all products induce the product topology and the product measure. Thus, the topology and the measure on $\mathcal{G}_{m,n} \times \mathcal{H}_m^V \times \mathcal{H}_n^W$ are defined. We use the notation $\text{Meas}(\cdot)$ for the measure on $\mathcal{G}_{m,n} \times \mathcal{H}_m^V \times \mathcal{H}_n^W$.

Now we can state the following three theorems that can be viewed as generalized version of Theorem 4.2, 4.3, and 4.4.

**Theorem A.2.** *Let $X \subset \mathcal{G}_{m,n} \times \mathcal{H}_m^V \times \mathcal{H}_n^W \backslash \mathcal{D}_{foldable}$ be measurable with finite measure. For any $\epsilon > 0$, there exists some $F \in \mathcal{F}_{GNN}$, such that*

$$Meas\left(\{(G, H) \in X : \mathbb{I}_{F(G,H)>1/2} \neq \Phi_{feas}(G, H)\}\right) < \epsilon,$$

*where $\mathbb{I}_\cdot$ is the indicator function, i.e., $\mathbb{I}_{F(G,H)>1/2} = 1$ if $F(G, H) > 1/2$ and $\mathbb{I}_{F(G,H)>1/2} = 0$ otherwise.*

**Theorem A.3.** *Let $X \subset \mathcal{G}_{m,n} \times \mathcal{H}_m^V \times \mathcal{H}_n^W \backslash \mathcal{D}_{foldable}$ be measurable with finite measure. For any $\epsilon, \delta > 0$, the followings hold:*

*(i) There exists some $F_1 \in \mathcal{F}_{GNN}$ such that*

$$Meas\left(\{(G, H) \in X : \mathbb{I}_{F_1(G,H)>1/2} \neq \mathbb{I}_{\Phi_{obj}(G,H) \in \mathbb{R}}\}\right) < \epsilon.$$

*(ii) There exists some $F_2 \in \mathcal{F}_{GNN}$ such that*

$$Meas\left(\left\{(G, H) \in X \cap \Phi_{obj}^{-1}(\mathbb{R}) : |F_2(G, H) - \Phi_{obj}(G, H)| > \delta\right\}\right) < \epsilon.$$

**Theorem A.4.** *Let $X \subset \mathcal{D}_{solu} \backslash \mathcal{D}_{foldable} \subset \mathcal{G}_{m,n} \times \mathcal{H}_m^V \times \mathcal{H}_n^W \backslash \mathcal{D}_{foldable}$ be measurable with finite measure. For any $\epsilon, \delta > 0$, there exists $F_W \in \mathcal{F}_{GNN}^W$ such that*

$$Meas\left(\{(G, H) \in X : \|F_W(G, H) - \Phi_{solu}(G, H)\| > \delta\}\right) < \epsilon.$$

The proof framework is similar to those in Chen et al. (2022), and consists of two steps: i) show that measurability of the target mapping and apply Lusin's theorem to obtain a continuous mapping on a

compact domain; ii) use Stone-Weierstrass-type theorem to show the uniform approximation result of graph neural networks.

We first prove that $\Phi_{\text{feas}}$ and $\Phi_{\text{obj}}$ are both measurable in the following two lemmas. The optimal solution mapping $\Phi_{\text{solu}}$ will be defined rigorously and proved as measurable in Section C.

**Lemma A.5.** *The feasibility mapping $\Phi_{feas} : \mathcal{G}_{m,n} \times \mathcal{H}_m^V \times \mathcal{H}_n^W \to \{0, 1\}$ is measurable.*

*Proof.* It suffices to prove that $\Phi_{\text{feas}}^{-1}(1)$ is measurable, and the proof is almost the same as the that of Chen et al. (2022, Lemma F.1). The difference is that we should consider any fixed $\tau \in \{0, 1\}^n$. Assuming $\tau = (0, \dots, 0, 1, \dots, 1)$ where 0 and 1 appear for $k$ and $n - k$ times respectively without loss of generality, we can replace $\mathbb{R}^n$ and $\mathbb{Q}^n$ in the proof of Chen et al. (2022, Lemma F.1) by $\mathbb{R}^k \times \mathbb{Z}^{n-k}$ and $\mathbb{Q}^k \times \mathbb{Z}^{n-k}$ to get the proof in the MILP setting. $\qquad\square$

**Lemma A.6.** *The optimal objective value mapping $\Phi_{obj} : \mathcal{G}_{m,n} \times \mathcal{H}_m^V \times \mathcal{H}_n^W \to \mathbb{R} \cup \{\infty, -\infty\}$ is measurable.*

*Proof.* Consider any $\phi \in \mathbb{R}$, $\Phi_{\text{obj}}(G, H) < \phi$ if and only if for the MILP problem associated to $(G, H)$, there exists a feasible solution $x$ and some $r \in \mathbb{N}_+$ such that $c^\top x \le \phi - 1/r$. The rest of the proof can be done using the same techniques as in the proofs of Chen et al. (2022, Lemma F.1 and Lemma F.2), with the difference pointed in the proof of Lemma A.5. $\qquad\square$

The Lusin's theorem, stated as follows, guarantees that any measurable function can be constricted on a compact such that only a small portion of domain is excluded and that the resulting function is continuous. The Lusin's theorem also applies for mappings defined on domains in $\mathcal{G}_{m,n} \times \mathcal{H}_m^V \times \mathcal{H}_n^W$. This is because that $\mathcal{G}_{m,n} \times \mathcal{H}_m^V \times \mathcal{H}_n^W$ is isomorphic to the disjoint union of finitely many Euclidean spaces.

**Theorem A.7** (Lusin's theorem (Evans & Garzepy, 2018, Theorem 1.14))**.** *Let $\mu$ be a Borel regular measure on $\mathbb{R}^n$ and let $f : \mathbb{R}^n \to \mathbb{R}^m$ be $\mu$-measurable. Then for any $\mu$-measurable $X \subset \mathbb{R}^n$ with $\mu(X) < \infty$ and any $\epsilon > 0$, there exists a compact set $E \subset X$ with $\mu(X \backslash E) < \epsilon$, such that $f|_E$ is continuous.*

The main tool in the proof of Theorem A.2 and Theorem A.3 is the universal approximation property of $\mathcal{F}_{\text{GNN}}$ stated below. Notice that the separation power of GNNs is the same as that of WL test. Theorem A.8 guarantees that GNNs can approximate any continuous function on a compact set whose separation power is less than or equal to that of the WL test, which can be proved using the Stone-Weierstrass theorem (Rudin, 1991, Section 5.7).

**Theorem A.8** (Theorem 4.3 in Chen et al. (2022))**.** *Let $X \subset \mathcal{G}_{m,n} \times \mathcal{H}_m^V \times \mathcal{H}_n^W$ be a compact set. For any $\Phi \in \mathcal{C}(X, \mathbb{R})$ satisfying*

$$\Phi(G, H) = \Phi(\hat{G}, \hat{H}), \quad \forall\, (G, H) \sim (\hat{G}, \hat{H}), \tag{A.1}$$

*and any $\epsilon > 0$, there exists $F \in \mathcal{F}_{GNN}$ such that*

$$\sup_{(G,H) \in X} |\Phi(G, H) - F(G, H)| < \epsilon. \tag{A.2}$$

Let us remark that the proof of Chen et al. (2022, Theorem 4.3) does not depend on the specific choice of the feature spaces $\mathcal{H}^V$ and $\mathcal{H}^W$, which only requires that $\mathcal{H}^V$ and $\mathcal{H}^W$ are both topological linear spaces. Therefore, Theorem A.8 works for MILP-graphs with additional vertices features.

For the sake of completeness, we outline the proof of Chen et al. (2022, Theorem 4.3): Let $\pi : \mathcal{G}_{m,n} \times \mathcal{H}_m^V \times \mathcal{H}_n^W \to \mathcal{G}_{m,n} \times \mathcal{H}_m^V \times \mathcal{H}_n^W / \sim$ be the quotient. Then for any continuous function $F : \mathcal{G}_{m,n} \times \mathcal{H}_m^V \times \mathcal{H}_n^W \to \mathbb{R}$, there exists a unique $\tilde{F} : \mathcal{G}_{m,n} \times \mathcal{H}_m^V \times \mathcal{H}_n^W / \sim \to \mathbb{R}$ such that $F = \tilde{F} \circ \pi$. Note that $\tilde{\mathcal{F}}_{\text{GNN}} := \{\tilde{F} : F \in \mathcal{F}_{\text{GNN}}\}$ separates points in $\mathcal{G}_{m,n} \times \mathcal{H}_m^V \times \mathcal{H}_n^W / \sim$ by Theorem 3.1. By the Stone-Weierstrass theorem, there exists $\tilde{F} \in \tilde{\mathcal{F}}_{\text{GNN}}$ with $\sum_{(G,H) \in X} |\tilde{\Phi}(\pi(G, H)) - (\pi(G, H))| < \epsilon$, which then implies (A.2).

Similar universal approximation results for GNNs and invariant target mappings can also be found in earlier literature; see e.g. Azizian & Lelarge (2021); Geerts & Reutter (2022). With all the preparation above, we can then proceed to present the proof of Theorem A.2.

For applying Theorem A.8, one need to verify that the separation power of $\Phi_{\text{feas}}$ and $\Phi_{\text{obj}}$ are bounded by that of WL test for unfoldable MILP instances, i.e., the condition (A.1). For any $(G, H), (\hat{G}, \hat{H}) \in \mathcal{G}_{m,n} \times \mathcal{H}_m^V \times \mathcal{H}_n^W \backslash \mathcal{D}_{\text{foldable}}$, since they are unfoldable, WL test should generate discrete coloring for them if there is no collision. If we further assume that they cannot be distinguished by WL test, then their stable discrete coloring must be identical (up to some permutation probably), which implies that they must be isomorphic. Therefore, we have the following lemma:

**Lemma A.9.** *For any $(G, H), (\hat{G}, \hat{H}) \in \mathcal{G}_{m,n} \times \mathcal{H}_m^V \times \mathcal{H}_n^W \backslash \mathcal{D}_{\text{foldable}}$, it holds that $(G, H) \sim (\hat{G}, \hat{H})$ if and only if $(G, H)$ and $(\hat{G}, \hat{H})$ are isomorphic, i.e., $(\sigma_V, \sigma_W) * (G, H) = (\hat{G}, \hat{H})$ for some $\sigma_V \in S_m$ and $\sigma_W \in S_n$.*

Let us remark that Lemma A.9 is true for any vertex feature spaces because the following proof does not assume the structure of the feature spaces.

*Proof.* It's trivial that the isomorphism of $(G, H)$ and $(\hat{G}, \hat{H})$ implies $(G, H) \sim (\hat{G}, \hat{H})$, and it suffices to show $(G, H) \sim (\hat{G}, \hat{H})$ implies the isomorphism.

Applying Algorithm 1 on $(G, H)$ and $(\hat{G}, \hat{H})$ with a common set of hash functions, we define the outcomes respectively:

$$\{\{C_i^{L,V}\}\}_{i=0}^m, \{\{C_j^{L,W}\}\}_{j=0}^n := \text{WL}_{\text{MILP}}((G, H), L),$$

$$\{\{\hat{C}_i^{L,V}\}\}_{i=0}^m, \{\{\hat{C}_j^{L,W}\}\}_{j=0}^n := \text{WL}_{\text{MILP}}((\hat{G}, \hat{H}), L).$$

Then $(G, H) \sim (\hat{G}, \hat{H})$ can be written as

$$\{\{C_i^{L,V}\}\}_{i=0}^m = \{\{\hat{C}_i^{L,V}\}\}_{i=0}^m, \text{ and } \{\{C_j^{L,W}\}\}_{j=0}^n = \{\{\hat{C}_j^{L,W}\}\}_{j=0}^n \tag{A.3}$$

for all $L \in \mathbb{N}$ and all hash functions. We define hash functions used in this proof.

- We take hash functions $\text{HASH}_{0,V}, \text{HASH}_{0,W}$ such that they are injective on the following domain:

$$\{h_i^V, h_j^W, \hat{h}_i^V, \hat{h}_j^W : 1 \leq i \leq m, 1 \leq j \leq n\}.$$

Note that $\{\cdot\}$ means a set, but not a multiset. The above set collects all possible values of the vertex features in the given two graphs $(G, H)$ and $(\hat{G}, \hat{H})$. Such hash functions exist because the number of vertices in the two graphs must be finite.

- For $l \in \{1, 2, \cdots, L\}$, we define a set that collects all possible vertex colors at the $l-1$-th iteration,

$$\boldsymbol{C}_{l-1} := \{C_i^{l-1,V}, C_j^{l-1,W}, \hat{C}_i^{l-1,V}, \hat{C}_j^{l-1,W} : 1 \leq i \leq m, 1 \leq j \leq n\}.$$

Note that $|\boldsymbol{C}_{l-1}|$ is finite. The two sets $\{\text{HASH}'_{l,V}(C) : C \in \boldsymbol{C}_{l-1}\}$ and $\{\text{HASH}'_{l,W}(C) : C \in \boldsymbol{C}_{l-1}\}$ can be linearly independent for some hash functions $\text{HASH}'_{l,V}, \text{HASH}'_{l,W}$. This is because that one can take the output spaces of $\text{HASH}'_{l,V}, \text{HASH}'_{l,W}$ as linear spaces of dimension greater than $|\boldsymbol{C}_{l-1}|$. Given such functions $\text{HASH}'_{l,V}, \text{HASH}'_{l,W}$, we consider the following set

$$\left\{ \left( C_i^{l-1,V}, \sum_{j=1}^n E_{i,j} \text{HASH}'_{l,W}\left( C_j^{l-1,W} \right) \right) : 1 \leq i \leq m \right\}$$

$$\cup \left\{ \left( \hat{C}_i^{l-1,V}, \sum_{j=1}^n \hat{E}_{i,j} \text{HASH}'_{l,W}\left( \hat{C}_j^{l-1,W} \right) \right) : 1 \leq i \leq m \right\}$$

$$\cup \left\{ \left( C_j^{l-1,W}, \sum_{i=1}^n E_{i,j} \text{HASH}'_{l,V}\left( C_i^{l-1,V} \right) \right) : 1 \leq j \leq n \right\}$$

$$\cup \left\{ \left( \hat{C}_j^{l-1,W}, \sum_{i=1}^n \hat{E}_{i,j} \text{HASH}'_{l,V}\left( \hat{C}_i^{l-1,V} \right) \right) : 1 \leq j \leq n \right\},$$

which is a finite set given any two graphs $(G, H)$ and $(\hat{G}, \hat{H})$. Thus, one can pick hash functions $\text{HASH}_{l,V}, \text{HASH}_{l,W}$ that are injective on the above set.

Due to the statement assumption that $(G, H)$ and $(\hat{G}, \hat{H})$ are both unfoldable and the definition of the above hash functions, one can obtain that

$$
\begin{aligned}
C_i^{L,V} \neq C_{i'}^{L,V}, \ \forall \, i \neq i', \quad & C_j^{L,W} \neq C_{j'}^{L,W}, \ \forall \, j \neq j', \\
\hat{C}_i^{L,V} \neq \hat{C}_{i'}^{L,V}, \ \forall \, i \neq i', \quad & \hat{C}_j^{L,W} \neq \hat{C}_{j'}^{L,W}, \ \forall \, j \neq j',
\end{aligned}
\tag{A.4}
$$

for some $L \in \mathbb{N}$. In another word, in either $(G, H)$ or $(\hat{G}, \hat{H})$, each vertex has an unique color. Combining (A.3) and (A.4), we have that there exists unique $\sigma_V \in S_m$ and $\sigma_W \in S_n$ such that

$$
\hat{C}_i^{L,V} = C_{\sigma_V(i)}^{L,V}, \quad \hat{C}_j^{L,W} = C_{\sigma_W(j)}^{L,W}, \quad \forall \, i = 1, 2, \ldots, m, j = 1, 2, \ldots, n.
$$

In addition, the vertices with the same colors much have the same neighbour information, including the colors of the nerighbourhood and weights of the associated edges. Rigorously, since $\mathrm{HASH}_{L,V}, \mathrm{HASH}_{L,W}$ are both injective, one will obtain that

$$
\hat{C}_i^{L-1,V} = C_{\sigma_V(i)}^{L-1,V}, \quad \hat{C}_j^{L-1,W} = C_{\sigma_W(j)}^{L-1,W},
$$

$$
\sum_{j=1}^{n} \hat{E}_{i,j} \mathrm{HASH}'_{L,W} \left( \hat{C}_j^{L-1,W} \right) = \sum_{j=1}^{n} E_{\sigma_V(i),j} \mathrm{HASH}'_{L,W} \left( C_j^{L-1,W} \right),
$$

$$
\sum_{i=1}^{n} \hat{E}_{i,j} \mathrm{HASH}'_{L,V} \left( \hat{C}_i^{L-1,V} \right) = \sum_{i=1}^{n} E_{i,\sigma_W(j)} \mathrm{HASH}'_{L,V} \left( C_i^{L-1,V} \right).
$$

Now let us consider the second line of the above equations,

$$
\begin{aligned}
\sum_{j=1}^{n} \hat{E}_{i,j} \mathrm{HASH}'_{L,W} \left( \hat{C}_j^{L-1,W} \right) &= \sum_{j=1}^{n} E_{\sigma_V(i),j} \mathrm{HASH}'_{L,W} \left( C_j^{L-1,W} \right) \\
&= \sum_{j=1}^{n} E_{\sigma_V(i),\sigma_W(j)} \mathrm{HASH}'_{L,W} \left( C_{\sigma_W(j)}^{L-1,W} \right) \\
&= \sum_{j=1}^{n} E_{\sigma_V(i),\sigma_W(j)} \mathrm{HASH}'_{L,W} \left( \hat{C}_j^{L-1,W} \right).
\end{aligned}
$$

Therefore, it holds that

$$
\sum_{j=1}^{n} \left( \hat{E}_{i,j} - E_{\sigma_V(i),\sigma_W(j)} \right) \mathrm{HASH}'_{L,W} \left( \hat{C}_j^{L-1,W} \right) = 0.
$$

This implies that $\hat{E}_{i,j} = E_{\sigma_V(i),\sigma_W(j)}$ because $\{\mathrm{HASH}'_{L,W}(C) : C \in \boldsymbol{C}_{L-1}\}$ are linearly independent. Then one can conclude that $(\sigma_V, \sigma_W) * (G, H) = (\hat{G}, \hat{H})$. $\qquad \square$

Now we can proceed to present the proof of Theorem A.2.

*Proof of Theorem A.2.* According to Lemma A.5, the mapping $\Phi_{\mathrm{feas}} : \mathcal{G}_{m,n} \times \mathcal{H}_m^V \times \mathcal{H}_n^W \to \{0, 1\}$ is measurable. By Lusin's theorem, there exists a compact $E \subset X \subset \mathcal{G}_{m,n} \times \mathcal{H}_m^V \times \mathcal{H}_n^W \backslash \mathcal{D}_{\mathrm{foldable}}$ such that $\Phi_{\mathrm{feas}}|_E$ is continuous and that $\mathrm{Meas}(X \backslash E) < \epsilon$. For any $(G, H), (\hat{G}, \hat{H}) \in E$, if $(G, H) \sim (\hat{G}, \hat{H})$, Lemma A.9 guarantees that $\Phi_{\mathrm{feas}}(G, H) = \Phi_{\mathrm{feas}}(\hat{G}, \hat{H})$. Then using Theorem A.8, one can conclude that there exists $F \in \mathcal{F}_{\mathrm{GNN}}$ with

$$
\sup_{(G,H) \in E} |F(G, H) - \Phi_{\mathrm{feas}}(G, H)| < \frac{1}{2}.
$$

This implies that

$$
\mathrm{Meas} \left( \left\{ (G, H) \in X : \mathbb{I}_{F(G,H) > 1/2} \neq \Phi_{\mathrm{feas}}(G, H) \right\} \right) \leq \mathrm{Meas}(X \backslash E) < \epsilon.
$$

$\qquad \square$

Similar techniques can be used to prove a sequence of results:

*Proof of Theorem 4.2.* $\mathcal{D}$ is finite and compact and $\Phi_{\text{feas}}|_{\mathcal{D}}$ is continuous. The rest of the proof follows the same argument as in the proof of Theorem A.2, using Lemma A.9 and Theorem A.8. $\quad\square$

*Proof of Theorem A.3.* $\Phi_{\text{obj}}$ is measurable by Lemma A.6, which implies that

$$\mathbb{I}_{\Phi_{\text{obj}}(\cdot,\cdot)\in\mathbb{R}} : \mathcal{G}_{m,n} \times \mathcal{H}_m^V \times \mathcal{H}_n^W \to \{0, 1\},$$

is also measurable. Then (i) and (ii) can be proved using the same argument as in the proof of Theorem A.2 for $X$ and $\mathbb{I}_{\Phi_{\text{obj}}(\cdot,\cdot)\in\mathbb{R}}$, as well as $X \cap \Phi_{\text{obj}}^{-1}(\mathbb{R})$ and $\Phi_{\text{obj}}$, respectively. $\quad\square$

*Proof of Theorem 4.3.* $\mathcal{D}$ is compact and $\mathbb{I}_{\Phi_{\text{obj}}(\cdot,\cdot)\in\mathbb{R}}$ restricted on $\mathcal{D}$ is continuous. In addition, as long as $\mathcal{D} \cap \Phi_{\text{obj}}^{-1}(\mathbb{R})$ is nonempty, it is compact and $\Phi_{\text{obj}}$ restricted on $\mathcal{D} \cap \Phi_{\text{obj}}^{-1}(\mathbb{R})$ is continuous. Then one can use Lemma A.9 and Theorem A.8 to obtain the desired results. $\quad\square$

Now we consider the optimal solution mapping $\Phi_{\text{solu}}$ and Theorem A.4, for proving which one needs a universal approximation result of $\mathcal{F}_{\text{GNN}}^W$ for equivariant functions. For stating the theorem rigorously, we define another equivalence relationship on $\mathcal{G}_{m,n} \times \mathcal{H}_m^V \times \mathcal{H}_n^W$: $(G, H) \overset{W}{\sim} (\hat{G}, \hat{H})$ if $(G, H) \sim (\hat{G}, \hat{H})$ and in addition, $C_j^{L,W} = \hat{C}_j^{L,W}$, $\forall\, j \in \{1, 2, \ldots, n\}$, for any $L \in \mathbb{N}$ and any hash functions.

The universal approximation result of $\mathcal{F}_{\text{GNN}}^W$ is stated as follows, which guarantees that $\mathcal{F}_{\text{GNN}}^W$ can approximate any continuous equivariant function on a compact set whose separation power is less than or equal to that of the WL test, and is based on a generalized Stone-Weierstrass theorem established in Azizian & Lelarge (2021).

**Theorem A.10** (Theorem E.1 in Chen et al. (2022)). *Let $X \subset \mathcal{G}_{m,n} \times \mathcal{H}_m^V \times \mathcal{H}_n^W$ be a compact subset that is closed under the action of $S_m \times S_n$. Suppose that $\Phi \in \mathcal{C}(X, \mathbb{R}^n)$ satisfies the followings:*

(i) *For any $\sigma_V \in S_m$, $\sigma_W \in S_n$, and $(G, H) \in X$, it holds that*

$$\Phi\left((\sigma_V, \sigma_W) * (G, H)\right) = \sigma_W(\Phi(G, H)).$$

(ii) $\Phi(G, H) = \Phi(\hat{G}, \hat{H})$ *holds for all $(G, H), (\hat{G}, \hat{H}) \in X$ with $(G, H) \overset{W}{\sim} (\hat{G}, \hat{H})$.*

(iii) *Given any $(G, H) \in X$ and any $j, j' \in \{1, 2, \ldots, n\}$, if $C_j^{l,W} = C_{j'}^{l,W}$ holds for any $l \in \mathbb{N}$ and any choices of hash functions, then $\Phi(G, H)_j = \Phi(G, H)_{j'}$.*

*Then for any $\epsilon > 0$, there exists $F_W \in \mathcal{F}_{GNN}^W$ such that*

$$\sup_{(G,H)\in X} \|\Phi(G, H) - F_W(G, H)\| < \epsilon.$$

Similar to Theorem A.8, Theorem A.10 is also true for all topological linear spaces $\mathcal{H}^V$ and $\mathcal{H}^W$. We refer to Azizian & Lelarge (2021); Geerts & Reutter (2022) for other results on the closure of equivariant GNNs. Now we can prove Theorem A.4 and Corollary 4.4.

*Proof of Theorem A.4.* We can assume that $X \subset \mathcal{D}_{\text{solu}}\backslash\mathcal{D}_{\text{foldable}} \subset \mathcal{G}_{m,n} \times \mathcal{H}_m^V \times \mathcal{H}_n^W\backslash\mathcal{D}_{\text{foldable}}$ is close under the action of $S_m \times S_n$, since otherwise one can replace $X$ with $\bigcup_{(\sigma_V,\sigma_W)\in S_m\times S_n}(\sigma_V, \sigma_W) * X$. The solution mapping $\Phi_{\text{solu}} : \mathcal{D}_{\text{solu}}\backslash\mathcal{D}_{\text{foldable}}$ is measurable by Lemma C.5. According to Lusin's theorem, there exists a compact $E' \subset X \subset \mathcal{D}_{\text{solu}}\backslash\mathcal{D}_{\text{foldable}}$ with $\text{Meas}(X\backslash E') < \epsilon/|S_m \times S_n|$ such that $\Phi_{\text{solu}}|_{E'}$ is continuous. Let us set

$$E = \bigcap_{(\sigma_V,\sigma_W)\in S_m\times S_n} (\sigma_V, \sigma_W) * E' \subset X,$$

which is compact and is closed under the action of $S_m \times S_n$. Furthermore, it holds that

$$\text{Meas}(X\backslash E) = \text{Meas}\left(X\backslash \bigcap_{(\sigma_V,\sigma_W)\in S_m\times S_n} (\sigma_V, \sigma_W) * E'\right)$$

$$\leq \sum_{(\sigma_V, \sigma_W) \in S_m \times S_n} \text{Meas}(X \backslash (\sigma_V, \sigma_W) * E')$$
$$= |S_m \times S_n| \cdot \text{Meas}(X \backslash E')$$
$$< \epsilon.$$

For any $(G, H), (\hat{G}, \hat{H}) \in E$, if $(G, H) \overset{W}{\sim} (\hat{G}, \hat{H})$, similar to Lemma A.9, we know that there exists $\sigma_V \in S_m$ such that $(\sigma_V, \text{Id}) * (G, H) = (\hat{G}, \hat{H})$. Then by the construction of $\Phi_{\text{solu}}$ in Section C, one has that $\Phi_{\text{solu}}(G, H) = \Phi_{\text{solu}}(\hat{G}, \hat{H}) \in \mathbb{R}^n$. Condition (i) in Theorem A.10 is satisfied by the definition of $\Phi_{\text{solu}}$ (see Section C) and Condition (iii) in Theorem A.10 follows from the fact that WL test yields discrete coloring for any graph in $\mathcal{G}_{m,n} \times \mathcal{H}_m^V \times \mathcal{H}_n^W \backslash \mathcal{D}_{\text{foldable}}$. Then applying Theorem A.10, one can conclude that for any $\delta > 0$, there exists $F_W \in \mathcal{F}_{\text{GNN}}^W$ with

$$\sup_{(G,H) \in E} \|F_W(G, H) - \Phi_{\text{solu}}(G, H)\| < \delta.$$

Therefore, it holds that

$$\text{Meas}\left(\{(G, H) \in X : \|F_W(G, H) - \Phi_{\text{solu}}(G, H)\| > \delta\}\right) \leq \text{Meas}(X \backslash E) < \epsilon,$$

which completes the proof. $\qquad\square$

*Proof of Theorem 4.4.* Without loss of generality, we can assume that the finite dataset $\mathcal{D} \subset \mathcal{D}_{\text{solu}} \backslash \mathcal{D}_{\text{foldable}}$ is closed under the action of $S_m \times S_n$. Otherwise, we can replace $\mathcal{D}$ by $\bigcup_{(\sigma_V, \sigma_W) \in S_m \times S_n} (\sigma_V, \sigma_W) * \mathcal{D}$.

Note that $\mathcal{D}$ is compact and $\Phi_{\text{solu}}|_\mathcal{D}$ is continuous. The rest of the proof can be done by using similar techniques as in the proof of Theorem A.4, with Theorem A.10. $\qquad\square$

# B  PROOFS FOR SECTION 5

We collect the proof of Theorem 5.1, Theorem 5.2, and Theorem 5.3 in this section. Before we proceed, we remark that the following concepts are modified as follows in this section due to the random feature technique:

- The space of MILP-graph. Each vertex in the MILP-graph is equipped with an additional feature and the entire random vector $\omega$ is sampled from $\omega \in \Omega := [0, 1]^m \times [0, 1]^n$. More specifically, we write:

$$\omega := (\omega^V, \omega^W) := (\omega_1^V, \cdots, \omega_m^V, \omega_1^W, \cdots, \omega_n^W).$$

Equipped with a random feature, the new vertex features are defined with:

$$h_i^{V,R} := (h_i^V, \omega_i^V), \ \ h_j^{W,R} := (h_j^W, \omega_j^W), \quad \forall \, 1 \leq i \leq m, 1 \leq j \leq n.$$

The corresponding vertex feature spaces are defined with:

$$\mathcal{H}_m^{V,R} := \mathcal{H}_m^V \times [0, 1]^m = (\mathcal{H}^V \times [0, 1])^m, \quad \mathcal{H}_n^{W,R} := \mathcal{H}_n^W \times [0, 1]^n = (\mathcal{H}^W \times [0, 1])^n.$$

The corresponding graph space is $\mathcal{G}_{m,n} \times \mathcal{H}_m^{V,R} \times \mathcal{H}_n^{W,R}$, and it is isomorphic to the space defined in the main text:

$$\mathcal{G}_{m,n} \times \mathcal{H}_m^V \times \mathcal{H}_n^W \times \Omega \cong \mathcal{G}_{m,n} \times \mathcal{H}_m^{V,R} \times \mathcal{H}_n^{W,R}. \tag{B.1}$$

Note that the two spaces are actually the same, merely defined in different way. In the main text, we adopt the former due to its simplicity. In the proof, we may either use the former or the latter according to the context.

- Topology and measure. We equip $\Omega$ with the standard topology and the Lebesgue measure. Then the overall space $\mathcal{G}_{m,n} \times \mathcal{H}_m^V \times \mathcal{H}_n^W \times \Omega$ can be naturally equipped with the product topology and the product measure.

- **Key mappings.** On the space (with random features) $\mathcal{G}_{m,n} \times \mathcal{H}_m^V \times \mathcal{H}_n^W \times \Omega$, the three key mappings can be defined with

$$
\begin{aligned}
\Phi_{\text{feas}}(G, H, \omega) &:= \Phi_{\text{feas}}(G, H), \\
\Phi_{\text{obj}}(G, H, \omega) &:= \Phi_{\text{obj}}(G, H), \\
\Phi_{\text{solu}}(G, H, \omega) &:= \Phi_{\text{solu}}(G, H),
\end{aligned}
$$

for all $\omega \in \Omega$. This is due to the fact that the feasibility, objective and optimal solution only depend on the MILP instance and they are independent of the random features appended.

- **Invariance and equivariance.** For any permutations $\sigma_V \in S_m$ and $\sigma_W \in S_n$, $(\sigma_V, \sigma_W) * (G, H, \omega)$ denotes the reordered MILP-graph (equipped with random features). We say a function $F_R$ is permutation invariant if

$$
F_R((\sigma_V, \sigma_W) * (G, H, \omega)) = F_R(G, H, \omega),
$$

for all $\sigma_V \in S_m$, $\sigma_W \in S_n$, and $(G, H, \omega) \in \mathcal{G}_{m,n} \times \mathcal{H}_m^{V,R} \times \mathcal{H}_n^{W,R}$. And we say a function $F_{W,R}$ is permutation equivariant if

$$
F_{W,R}((\sigma_V, \sigma_W) * (G, H, \omega)) = \sigma_W(F_{W,R}(G, H, \omega)),
$$

for all $\sigma_V \in S_m$, $\sigma_W \in S_n$, and $(G, H, \omega) \in \mathcal{G}_{m,n} \times \mathcal{H}_m^{V,R} \times \mathcal{H}_n^{W,R}$. It's clear that $\Phi_{\text{feas}}$ and $\Phi_{\text{obj}}$ are permutation invariant, and $\Phi_{\text{solu}}$ is permutation equivariant. Furthermore, it holds that any $F_R \in \mathcal{F}_{\text{GNN}}^R$ is invariant and any $F_{W,R} \in \mathcal{F}_{\text{GNN}}^{W,R}$ is equivariant.

- **The WL test.** In the WL test, vertex features $h_i^V, h_j^W$ are replaced with $h_i^{V,R}, h_j^{W,R}$, respectively, and the corresponding vertex colors are denoted by $C_i^{l,V,R}, C_j^{0,W,R}$ in the $l$-th iteration.

$$
\text{Initialization:} \quad C_i^{0,V,R} = \text{HASH}_{0,V}(h_i^{V,R}), \quad C_j^{0,W,R} = \text{HASH}_{0,W}(h_j^{W,R}).
$$

$$
\text{For } l = 1, 2, \cdots, L, \quad C_i^{l,V,R} = \text{HASH}_{l,V}\left(C_i^{l-1,V,R}, \sum_{j=1}^n E_{i,j} \text{HASH}_{l,W}'\left(C_j^{l-1,W,R}\right)\right).
$$

$$
\text{For } l = 1, 2, \cdots, L, \quad C_j^{l,W,R} = \text{HASH}_{l,W}\left(C_j^{l-1,W,R}, \sum_{i=1}^m E_{i,j} \text{HASH}_{l,V}'\left(C_i^{l-1,V,R}\right)\right).
$$

(B.2)

Note that the update form of (B.2) is equal to that of Algorithm 1, and the only difference is the initial vertex features.

- **The equivalence relationship.** Now let us define the equivalence relationships $\sim$ and $\overset{W}{\sim}$ on the space equipped with random features $\mathcal{G}_{m,n} \times \mathcal{H}_m^V \times \mathcal{H}_n^W \times \Omega$. Given two MILP-graphs with random features $(G, H, \omega)$ and $(\hat{G}, \hat{H}, \hat{\omega})$, the outcomes of the WL test are $\{\{C_i^{L,V,R}\}\}_{i=0}^m, \{\{C_j^{L,W,R}\}\}_{j=0}^n$ and $\{\{\hat{C}_i^{L,V,R}\}\}_{i=0}^m, \{\{\hat{C}_j^{L,W,R}\}\}_{j=0}^n$, respectively. We say $(G, H, \omega) \sim (\hat{G}, \hat{H}, \hat{\omega})$ if

$$
\{\{C_i^{L,V,R}\}\}_{i=0}^m = \{\{\hat{C}_i^{L,V,R}\}\}_{i=0}^m \text{ and } \{\{C_j^{L,W,R}\}\}_{j=0}^n = \{\{\hat{C}_j^{L,W,R}\}\}_{j=0}^n,
$$

for all for any $L \in \mathbb{N}$ and any hash functions. We say $(G, H, \omega) \overset{W}{\sim} (\hat{G}, \hat{H}, \hat{\omega})$ if $(G, H, \omega) \sim (\hat{G}, \hat{H}, \hat{\omega})$ and in addition, $C_j^{L,W,R} = \hat{C}_j^{L,W,R}$, $\forall j \in \{1, 2, \ldots, n\}$, for any $L \in \mathbb{N}$ and any hash functions.

- **Foldability.** Suppose the vertex colors corresponding to $(G, H, \omega)$ are $C_i^{l,V,R}, C_j^{0,W,R}$. Then we say $(G, H, \omega)$ is foldable if there exist $1 \leq i, i' \leq m$ or $1 \leq j, j' \leq n$ such that

$$
C_i^{l,V,R} = C_{i'}^{l,V,R}, \ i \neq i', \quad \text{or} \quad C_j^{l,W,R} = C_{j'}^{l,W,R}, \ j \neq j',
$$

for all $l \in \mathbb{N}$ and any hash functions.

By merely replacing $h_i^V, h_j^W$ with $h_i^{V,R}, h_j^{W,R}$, we define all the required preliminaries. According to the remarks following Theorem A.8, Lemma A.9, and Theorem A.10, we are able to directly apply these theorems and lemmas on the space $\mathcal{G}_{m,n} \times \mathcal{H}_m^{V,R} \times \mathcal{H}_n^{W,R}$, where $\mathcal{H}_m^{V,R}$ corresponds to $\mathcal{H}_m^V$ and $\mathcal{H}_n^{W,R}$ corresponds to $\mathcal{H}_n^W$.

*Proof of Theorem 5.1.* Define

$$\Omega_F = \{\omega \in \Omega : \exists\, i \neq i' \in \{1, 2, \dots, m\}, \text{ s.t. } \omega_i^V = \omega_{i'}^V,$$
$$\text{or } \exists\, j \neq j' \in \{1, 2, \dots, n\}, \text{ s.t. } \omega_j^W = \omega_{j'}^W \}. \tag{B.3}$$

Given any $(G, H) \in \mathcal{G}_{m,n} \times \mathcal{H}_m^V \times \mathcal{H}_n^W$, as long as $\omega \notin \Omega_F$, different vertices in $V$ or $W$ are equipped with different vertex features:

$$h_i^{V,R} \neq h_{i'}^{V,R}, \ \forall i \neq i', \quad h_j^{W,R} \neq h_{j'}^{W,R}, \ \forall j \neq j',$$

and consequently, each of the vertices possesses an unique initial color:

$$C_i^{0,V,R} \neq C_{i'}^{0,V,R}, \ \forall i \neq i', \quad C_j^{0,W,R} \neq C_{j'}^{0,W,R}, \ \forall j \neq j',$$

for some $\text{HASH}_{0,V}$ and $\text{HASH}_{0,W}$. In another word, although $(G, H)$ may be foldable, $(G, H, \omega)$ must be unfoldable as long as $\omega \notin \Omega_F$.

Therefore, for any $(G, H, \omega), (\hat{G}, \hat{H}, \hat{\omega}) \in \mathcal{G}_{m,n} \times \mathcal{H}_m^V \times \mathcal{H}_n^W \times (\Omega \backslash \Omega_F)$, it holds that both $(G, H, \omega)$ and $(\hat{G}, \hat{H}, \hat{\omega})$ are unfoldable. Applying Lemma A.9, we have $(G, H, \omega) \sim (\hat{G}, \hat{H}, \hat{\omega})$ if and only if $(\sigma_V, \sigma_W) * (G, H, \omega) = (\hat{G}, \hat{H}, \hat{\omega})$ for some $\sigma_V \in S_m$ and $\sigma_W \in S_n$. The invariance property of $\Phi_{\text{feas}}$ implies that

$$\Phi_{\text{feas}}(G, H, \omega) = \Phi_{\text{feas}}(\hat{G}, \hat{H}, \hat{\omega}), \quad \forall (G, H, \omega) \sim (\hat{G}, \hat{H}, \hat{\omega}) \in \mathcal{G}_{m,n} \times \mathcal{H}_m^V \times \mathcal{H}_n^W \times (\Omega \backslash \Omega_F).$$

In addition, the probability of $\omega \in \Omega_F$ is zero due to the uniform distribution of $\omega$ : $\mathbb{P}(\Omega_F) = 0$. For any $\epsilon > 0$, there exists a compact subset $\Omega_\epsilon \subset \Omega \backslash \Omega_F$ such that

$$\mathbb{P}(\Omega \backslash \Omega_\epsilon) = \mathbb{P}\left((\Omega \backslash \Omega_F) \backslash \Omega_\epsilon\right) < \epsilon.$$

Note that $\mathcal{D} \times \Omega_\epsilon$ is also compact and that $\Phi_{\text{feas}}$ is continuous on $\mathcal{D} \times \Omega_\epsilon$. The continuity of $\Phi_{\text{feas}}|_{\mathcal{D} \times \Omega_\epsilon}$ follows from the condition that $\mathcal{D}$ is a finite dataset and that the fact that any mapping with discrete finite domain is continuous (note that $\Phi_{\text{feas}}(G, H, \omega)$ is independent of $\omega \in \Omega_\epsilon$). Applying Theorem A.8 for $\Phi_{\text{feas}}$ and $\mathcal{D} \times \Omega_\epsilon \subset \mathcal{G}_{m,n} \times \mathcal{H}_m^{V,R} \times \mathcal{H}_n^{W,R}$, one can conclude the existence of $F^R \in \mathcal{F}_{\text{GNN}}^R$ with

$$\sup_{(G, H, \omega) \in \mathcal{D} \times \Omega_\epsilon} |F_R(G, H, \omega) - \Phi_{\text{feas}}(G, H, \omega)| < \frac{1}{2}.$$

Since $\Phi_{\text{feas}}$ is independent of the random features $\omega$, the above equation can be rewritten as

$$\sup_{(G, H, \omega) \in \mathcal{D} \times \Omega_\epsilon} |F_R(G, H, \omega) - \Phi_{\text{feas}}(G, H)| < \frac{1}{2}.$$

It thus holds for any $(G, H) \in \mathcal{D}$ that

$$\mathbb{P}\left(\mathbb{I}_{F_R(G,H) > 1/2} \neq \Phi_{\text{feas}}(G, H)\right) \leq \mathbb{P}(\Omega \backslash \Omega_\epsilon) < \epsilon.$$

$\square$

*Proof of Theorem 5.2.* The results can be obtained by applying similar techniques as in the proof of Theorem 5.1 for $\mathbb{I}_{\Phi_{\text{obj}}(\cdot, \cdot) \in \mathbb{R}}$ and $\Phi_{\text{obj}}$.

$\square$

*Proof of Theorem 5.3.* Without loss of generality, we can assume that $\mathcal{D}$ is closed under the action of $S_m \times S_n$; otherwise, we can use $\{(\sigma_V, \sigma_W) * (G, H) : (G, H) \in \mathcal{D}, \sigma_V \in S_m, \sigma_W \in S_n\}$ instead of $\mathcal{D}$. Let $\Omega_F \subset \Omega$ be the set defined in (B.3) which is clearly closed under the action of $S_m \times S_n$. There exists a compact $\Omega_\epsilon' \subset \Omega \backslash \Omega_F$ with $\mathbb{P}(\Omega \backslash \Omega_\epsilon') = \mathbb{P}((\Omega \backslash \Omega_F) \backslash \Omega_\epsilon') < \epsilon / |S_m \times S_n|$. Define

$$\Omega_\epsilon = \bigcap_{(\sigma_V, \sigma_W) \in S_m \times S_n} (\sigma_V, \sigma_W) * \Omega_\epsilon' \subset \Omega \backslash \Omega_F,$$

which is compact and closed under the action of $S_m \times S_n$. One has that

$$\mathbb{P}(\Omega \backslash \Omega_\epsilon) \leq \mathbb{P}\left(\Omega \backslash \bigcap_{(\sigma_V, \sigma_W) \in S_m \times S_n} (\sigma_V, \sigma_W) * \Omega_\epsilon'\right)$$

$$\leq \sum_{(\sigma_V, \sigma_W) \in S_m \times S_n} \mathbb{P}(\Omega \backslash (\sigma_V, \sigma_W) * \Omega'_\epsilon)$$

$$= |S_m \times S_n| \cdot \mathbb{P}(\Omega \backslash \Omega'_\epsilon)$$

$$< \epsilon.$$

In addition, $\mathcal{D} \times \Omega_\epsilon$ is compact in $\mathcal{G}_{m,n} \times (\mathcal{H}^V \times [0,1])^m \times (\mathcal{H}^W \times [0,1])^n$ and is closed under the action of $S_m \times S_n$.

We then verify the three conditions in Theorem A.10 for $\Phi_{\text{solu}}$ and $\mathcal{D} \times \Omega_\epsilon$. Condition (i) holds automatically by the definition of the optimal solution mapping. For Condition (ii), given any $(G, H, \omega)$ and $(\hat{G}, \hat{H}, \hat{\omega})$ in $\mathcal{D} \times \Omega_\epsilon$ with $(G, H, \omega) \overset{W}{\sim} (\hat{G}, \hat{H}, \hat{\omega})$, since graphs with vertex features in $\mathcal{D} \times \Omega_\epsilon \subset \mathcal{G}_{m,n} \times \mathcal{H}_n^V \times \mathcal{H}_n^W \times (\Omega \backslash \Omega_F)$ cannot be folded, similar to Lemma A.9, one can conclude that $(\hat{G}, \hat{H}, \hat{\omega}) = (\sigma_V, \text{Id}) * (G, H, \omega)$ for some $\sigma_V \in S_n$, which leads to $\Phi_{\text{solu}}(G, H) = \Phi_{\text{solu}}(\hat{G}, \hat{H})$. Condition (iii) also follows from the fact that any graph in $\mathcal{D} \times \Omega_\epsilon$ cannot be folded and WL test yields discrete coloring. Therefore, Theorem A.10 applies and there exists $F_{W,R} \in \mathcal{F}_{\text{GNN}}^{W,R}$ such that

$$\sup_{(G, H, \omega) \in \mathcal{D} \times \Omega_\epsilon} \| F_{W,R}(G, H, \omega) - \Phi_{\text{solu}}(G, H) \| < \delta,$$

which implies that

$$\mathbb{P}(\| F_{W,R}(G, H) - \Phi_{\text{solu}}(G, H) \| > \delta) \leq \mathbb{R}(\Omega \backslash \Omega_\epsilon) < \epsilon, \quad \forall (G, H) \in \mathcal{D}.$$

$\square$

**Remark B.1.** *A more deeper observation is that, Theorem 5.1, Theorem 5.2, and Theorem A.3 are actually true even if we only allow one message-passing layer in the GNN structure. This is because that the separation power of GNNs with one message-passing layer is the same as the WL test with one iteration (see Chen et al. (2022, Appendix C)), and that one iteration in WL test suffices to yield a discrete coloring for MILP-graphs in $\mathcal{G}_{m,n} \times \mathcal{H}_n^V \times \mathcal{H}_n^W \times (\Omega \backslash \Omega_F)$.*

## C  THE OPTIMAL SOLUTION MAPPING $\Phi_{\text{SOLU}}$

In this section, we define the equivariant optimal solution mapping $\Phi_{\text{solu}}$, and prove the measurability. The definition consists of several steps.

### C.1  THE SORTING MAPPING

We first define a sorting mapping:

$$\Phi_{\text{sort}} : \mathcal{G}_{m,n} \times \mathcal{H}_m^V \times \mathcal{H}_n^W \backslash \mathcal{D}_{\text{foldable}} \to S_n,$$

that returns a permutation on $\{w_1, w_2, \dots, w_n\}$, $\mathcal{D}_{\text{foldable}}$ is the collection of all foldable MILP problem as in Definition 4.1. This can be done via an order refinement procedure, similar to the WL test. The initial order as well as the order refinement are defined in the following several definitions.

**Definition C.1.** *We define total order on $\mathcal{H}^V$ and $\mathcal{H}^W$ using lexicographic order:*

(i) *For any $(b_i, \circ_i), (b_{i'}, \circ_{i'}) \in \mathcal{H}^V = \mathbb{R} \times \{\leq, =, \geq\}$, we say $(b_i, \circ_i) < (b_{i'}, \circ_{i'})$ if one of the followings holds:*

- $b_i < b_{i'}$.
- $b_i = b_{i'}$ *and* $\iota(\circ_i) < \iota(\circ_{i'})$, *where* $\iota(\leq) = -1$, $\iota(=) = 0$, *and* $\iota(\geq) = 1$.

(ii) *For any $(c_j, l_j, u_j, \tau_j), (c_{j'}, l_{j'}, u_{j'}, \tau_{j'}) \in \mathcal{H}^W = \mathbb{R} \times (\mathbb{R} \cup \{-\infty\}) \times (\mathbb{R} \cup \{+\infty\}) \times \{0, 1\}$, we say $(c_j, l_j, u_j, \tau_j) < (c_{j'}, l_{j'}, u_{j'}, \tau_{j'})$ if one of the followings holds:*

- $c_j < c_{j'}$.
- $c_j = c_{j'}$ *and* $l_j < l_{j'}$.
- $c_j = c_{j'}$, $l_j = l_{j'}$, *and* $u_j < u_{j'}$.

$$- c_j = c_{j'},\ l_j = l_{j'},\ u_j = u_{j'},\ \text{and}\ \tau_j < \tau_{j'}.$$

**Definition C.2.** *Let $X = \{\{x_1, x_2, \ldots, x_k\}\}$ and $X' = \{\{x'_1, x'_2, \ldots, x'_{k'}\}\}$ be two multisets whose elements are taken from the same totally-ordered set. Then we say $X \leq X'$ if $x_{\sigma(1)} x_{\sigma(2)} \cdots x_{\sigma(k)} \leq x'_{\sigma'(1)} x'_{\sigma'(2)} \cdots x'_{\sigma'(k')}$ in the sense of lexicographical order, where $\sigma \in S_k$ and $\sigma' \in S_{k'}$ are permutations such that $x_{\sigma(1)} \leq x_{\sigma(2)} \leq \cdots \leq x_{\sigma(k)}$ and $x'_{\sigma'(1)} \leq x'_{\sigma'(2)} \leq \cdots \leq x'_{\sigma'(k')}$.*

**Definition C.3.** *Suppose that $V = \{v_1, v_2, \ldots, v_n\}$ and $W = \{w_1, w_2, \ldots, w_n\}$ are already ordered, and that $E \in \mathbb{R}^{m \times n}$. The order refinement is defined lexicographicaly:*

(i) *For any $i, i' \in \{1, 2, \ldots, m\}$, we say that*

$$\left(v_i, \{\{(E_{i,j}, w_j) : E_{i,j} \neq 0\}\}\right) < \left(v_{i'}, \{\{(E_{i',j}, w_j) : E_{i',j} \neq 0\}\}\right),$$

*if one of the followings holds:*

- $v_i < v_{i'}$.
- $v_i = v_{i'}$ *and* $\{(E_{i,j}, w_j) : E_{i,j} \neq 0\} < \{(E_{i',j}, w_j) : E_{i',j} \neq 0\}$ *in the sense of Definition C.2 where $(E_{i,j}, w_j) < (E_{i',j'}, w_{j'})$ if and only if $E_{i,j} < E_{i',j'}$ or $E_{i,j} = E_{i',j'}$, $w_j < w_{j'}$.*

(ii) *For any $j, j' \in \{1, 2, \ldots, n\}$, we say that*

$$\left(w_j, \{\{(E_{i,j}, v_i) : E_{i,j} \neq 0\}\}\right) < \left(w_{j'}, \{\{(E_{i,j'}, v_i) : E_{i,j'} \neq 0\}\}\right),$$

*if one of the followings holds:*

- $w_j < w_{j'}$.
- $w_j = w_{j'}$ *and* $\{(E_{i,j}, v_i) : E_{i,j} \neq 0\} < \{(E_{i,j'}, v_i) : E_{i,j'} \neq 0\}$ *in the sense of Definition C.2 where $(E_{i,j}, v_i) < (E_{i',j'}, v_{i'})$ if and only if $E_{i,j} < E_{i',j'}$ or $E_{i,j} = E_{i',j'}$, $v_i < v_{i'}$.*

With the preparations above, we can now define $\Phi_{\text{sort}}$ in Algorithm 2.

---

**Algorithm 2** The sorting mapping $\Phi_{\text{sort}}$

---

**Require:** A weighted bipartite graph $G = (V \cup W, E) \in \mathcal{G}_{m,n}$, with vertex features $H = (h_1^V, h_2^V, \ldots, h_m^V, h_1^W, h_2^W, \ldots, h_n^W) \in \mathcal{H}_m^V \times \mathcal{H}_n^W$ such that $(G, H) \notin \mathcal{D}_{\text{foldable}}$.
1: Order $V$ and $W$ according to Definition C.1.
2: **for** $l = 1 : m + n$ **do**
3:     Refine the ordering on $V$ and $W$ according to Definition C.3.
4: **end for**
5: Return $\sigma_W \in S_n$ such that $w_{\sigma_W(1)} < w_{\sigma_W(2)} < \cdots < w_{\sigma_W(n)}$.

---

**Remark C.4.** *The output of Algorithm 2 is well-defined and is unique. This is because that we use unfoldable $(G, H)$ as input. Note that the order refinement in Definition C.3 is more strict than the color refinement in WL test. Therefore, after $m + n$ iterations, there is no $j \neq j' \in \{1, 2, \ldots, n\}$ with $w_j = w_{j'}$, since the WL test returns discrete coloring if no collisions for $(G, H) \notin \mathcal{D}_{\text{foldable}}$.*

The sorting mapping $\Phi_{\text{sort}}$ has some straightforward properties:

- $\Phi_{\text{sort}}$ is equivariant:

$$\Phi_{\text{sort}}((\sigma_V, \sigma_W) * (G, H)) = (\sigma_V, \sigma_W) * \Phi_{\text{sort}}(G, H) = \sigma_W \circ \Phi_{\text{sort}}(G, H),$$

for any $\sigma_V \in S_m$, $\sigma_W \in S_n$, and $(G, H) \in \mathcal{G}_{m,n} \times \mathcal{H}_m^V \times \mathcal{H}_n^W \backslash \mathcal{D}_{\text{foldable}}$. This is due to the fact that Definition C.3 defines the total order and its refinement solely depending on the vertex features that are independent of the input order.

- $\Phi_{\text{sort}}$ is measurable, where the range $S_n$ is equipped with the discrete measure. This is because that $\Phi_{\text{sort}}$ is defined via finitely many comparisons.

## C.2 THE OPTIMAL SOLUTION MAPPING

We then define the optimal solution mapping

$$\Phi_{\text{solu}} : \mathcal{D}_{\text{solu}} \backslash \mathcal{D}_{\text{foldable}} \to \mathbb{R}^n,$$

based on $\Phi_{\text{sort}}$, where

$$\mathcal{D}_{\text{solu}} = \left( \mathcal{G}_{m,n} \times \mathcal{H}_m^V \times \widetilde{\mathcal{H}}_n^W \right) \cap \Phi_{\text{feas}}^{-1}(1),$$

with $\widetilde{\mathcal{H}}_n^W = (\mathbb{R} \times \mathbb{R} \times \mathbb{R} \times \{0,1\})^n \subset \mathcal{H}_n^W$, is the collection of all feasible MILP problems in which every component in $l$ and $u$ is finite.

We have mentioned before that any MILP problem in $\mathcal{D}_{\text{solu}}$ admits at least one optimal solution. For any $(G,H) \in \mathcal{D}_{\text{solu}} \backslash \mathcal{D}_{\text{foldable}}$, we denote $X_{\text{solu}}(G,H) \in \mathbb{R}^n$ as the set of optimal solutions to the MILP problem associated to $(G,H)$. One can see that $X_{\text{solu}}(G,H)$ is compact since for every $(G,H) \in \mathcal{D}_{\text{solu}} \backslash \mathcal{D}_{\text{foldable}}$, both $l$ and $u$ are finite, which leads to the boundedness (and hence the compactness) of $X_{\text{solu}}(G,H)$.

Given any permutation $\sigma \in S_n$, let us define a total order on $\mathbb{R}^d$: $x \overset{\sigma}{\prec} x'$ if and only if $x_{\sigma(1)} x_{\sigma(2)} \cdots x_{\sigma(n)} < x'_{\sigma(1)} x'_{\sigma(2)} \cdots x'_{\sigma(n)}$ in the sense of lexicographic order. For any $(G,H) \in \mathcal{D}_{\text{solu}} \backslash \mathcal{D}_{\text{foldable}} \subset \mathcal{G}_{m,n} \times \mathcal{H}_m^V \times \mathcal{H}_n^W \backslash \mathcal{D}_{\text{foldable}}$, as in Section C.1, the permutation $\Phi_{\text{sort}}(G,H) \in S_n$ is well-defined. Then we define $\Phi_{\text{solu}}(G,H)$ is the smallest element in $X_{\text{solu}}(G,H)$ with respect to the order $\overset{\sigma}{\prec}$, where $\sigma = \Phi_{\text{sort}}(G,H)$. The existence and the uniqueness are true since $X_{\text{solu}}(G,H)$ is compact. More explicitly, the components of $\Phi_{\text{solu}}(G,H)$ can be determined recursively:

$$\Phi_{\text{solu}}(G,H)_{\sigma(1)} = \inf \left\{ x_{\sigma(1)} : x \in X_{\text{solu}}(G,H) \right\},$$

and

$$\Phi_{\text{solu}}(G,H)_{\sigma(j)} = \inf \left\{ x_{\sigma(j)} : x \in X_{\text{solu}}(G,H), \ x_{\sigma(j')} = \Phi_{\text{solu}}(G,H)_{\sigma(j)}, \ j' = 1,2,\ldots,j \right\},$$

for $j = 2,3,\ldots,n$. It follows from the equivariance of $\Phi_{\text{sort}}(G,H)$ and $X_{\text{solu}}(G,H)$ that $\Phi_{\text{solu}}(G,H)$ is also equivariant, i.e.,

$$\Phi_{\text{solu}}((\sigma_V, \sigma_W) * (G,H)) = \sigma_W(\Phi_{\text{solu}}(G,H)), \quad \forall \sigma_V \in S_m, \sigma_W \in S_n, (G,H) \in \mathcal{D}_{\text{solu}} \backslash \mathcal{D}_{\text{foldable}}.$$

We then show the measurability of $\Phi_{\text{solu}}$.

**Lemma C.5.** *The optimal solution mapping $\Phi_{solu} : \mathcal{D}_{solu} \backslash \mathcal{D}_{foldable} \to \mathbb{R}^n$ is measurable.*

*Proof.* It suffices to show that for any fixed $\circ \in \{\leq, =, \geq\}^m$, $\tau \in \{0,1\}^n$, and $\sigma \in S_n$, the mapping

$$\begin{aligned} \Phi_j : \iota^{-1}(\mathcal{D}_{\text{solu}} \backslash \mathcal{D}_{\text{foldable}}) \cap (\Phi_{\text{sort}} \circ \iota)^{-1}(\sigma) &\to & \mathbb{R}, \\ (A,b,c,l,u) &\mapsto & \Phi_{\text{solu}}(\iota(A,b,c,l,u))_{\sigma(j)}, \end{aligned}$$

is measurable for all $j \in \{1,2,\ldots,n\}$, where

$$\iota : \mathbb{R}^{m \times n} \times \mathbb{R}^m \times \mathbb{R}^n \times \mathbb{R}^n \times \mathbb{R}^n \to \mathcal{G}_{m,n} \times \mathcal{H}_m^V \times \mathcal{H}_n^W,$$

is the embedding map when $\circ$ and $\tau$ are fixed. Without loss of generality, we assume that $\circ = \{\leq, \ldots, \leq, =, \ldots, =, \geq, \ldots, \geq\}$ where $\leq$, $=$, and $\geq$ appear for $k_1$, $k_2 - k_1$, and $m - k_2$ times, respectively, and that $\tau = (0,\ldots,0,1,\ldots,1)$ where $0$ and $1$ appear for $k$ and $n-k$ times, respectively. Note that the domain of $\Phi_j$, i.e., $\iota^{-1}(\mathcal{D}_{\text{solu}} \backslash \mathcal{D}_{\text{foldable}}) \cap (\Phi_{\text{sort}} \circ \iota)^{-1}(\sigma)$ is measurable due to the measurability of $\Phi_{\text{sort}}$.

Define

$$\begin{aligned} V_{\text{feas}}(A,b,c,l,u,x) = \max \Bigg\{ & \max_{1 \leq i \leq k_1} \left( \sum_{j'=1}^n A_{i,j'} x_{j'} - b_i \right)_+, \ \max_{k_1 < i \leq k_2} \left| \sum_{j'=1}^n A_{i,j'} x_{j'} - b_i \right|, \\ & \max_{k_2 < i \leq m} \left( b_i - \sum_{j'=1}^n A_{i,j'} x_{j'} \right)_+, \ \max_{1 \leq j' \leq n} (l_{j'} - x_{j'})_+, \ \max_{1 \leq j' \leq n} (x_{j'} - u_{j'})_+ \Bigg\}, \end{aligned}$$

and

$$V_{\text{solu}}(A, b, c, l, u, x) = \max\left\{\left(c^\top x - \Phi_{\text{obj}}(\iota(A, b, c, l, u))\right)_+, V_{\text{feas}}(A, b, c, l, u, x)\right\},$$

for $(A, b, c, l, u) \in \mathbb{R}^{m \times n} \times \mathbb{R}^m \times \mathbb{R}^n \times \mathbb{R}^n \times \mathbb{R}^n$ and $x \in \mathbb{R}^k \times \mathbb{Z}^{n-k}$. It is clear that $V_{\text{feas}}$ is continuous and that $x \in \mathbb{R}^k \times \mathbb{Z}^{n-k}$ is an optimal solution to the problem $\iota(A, b, c, l, u)$ if and only if $V_{\text{solu}}(A, b, c, l, u, x) = 0$. In addition, $V_{\text{solu}}$ is measurable with respect to $(A, b, c, l, u)$ for any $x$, by the measurability of $\Phi_{\text{obj}}$ (see Lemma A.6) and continuity of $V_{\text{feas}}$, and is continuous with respect to $x$.

Then we proceed to prove that $\Phi_j$ is measurable by induction. We first consider the case that $j = 1$. For any $(A, b, c, l, u) \in \iota^{-1}(\mathcal{D}_{\text{solu}} \backslash \mathcal{D}_{\text{foldable}}) \cap (\Phi_{\text{sort}} \circ \iota)^{-1}(\sigma)$ and any $\phi \in \mathbb{R}$, the followings are equivalent:

- $\Phi_{\text{solu}}(\iota(A, b, c, l, u))_{\sigma(1)} < \phi$.

- $\inf\left\{x_{\sigma(1)} : x \in X_{\text{solu}}(\iota(A, b, c, l, u))\right\} < \phi$.

- There exist $r \in \mathbb{N}_+$ and $x \in X_{\text{solu}}(\iota(A, b, c, l, u))$, such that $x_{\sigma(1)} \leq \phi - 1/r$.

- There exists $r \in \mathbb{N}_+$, for any $r' \in \mathbb{N}_+$, $x_{\sigma(1)} \leq \phi - 1/r$ holds for some $x \in \mathbb{Q}^k \times \mathbb{Z}^{n-k}$ satisfying $V_{\text{solu}}(A, b, c, l, u, x) < 1/r'$.

Therefore, it holds that

$$\Phi_1^{-1}(-\infty, \phi) = \iota^{-1}(\mathcal{D}_{\text{solu}} \backslash \mathcal{D}_{\text{foldable}}) \cap (\Phi_{\text{sort}} \circ \iota)^{-1}(\sigma) \cap \bigcup_{r \in \mathbb{N}_+} \bigcap_{r' \in \mathbb{N}_+} \bigcup_{x \in \mathbb{Q}^k \times \mathbb{Z}^{n-k}, \, x_{\sigma(1)} \leq \phi - 1/r}$$

$$\left\{(A, b, c, l, u) \in \mathbb{R}^{m \times n} \times \mathbb{R}^m \times \mathbb{R}^n \times \mathbb{R}^n \times \mathbb{R}^n : V_{\text{solu}}(A, b, c, l, u, x) < \frac{1}{r'}\right\},$$

is measurable, which implies the measurability of $\Phi_1$

Then we assume that $\Phi_1, \ldots, \Phi_{j-1}$ $(j \geq 2)$ are all measurable and show that $\Phi_j$ is also measurable. Define

$$V_{\text{solu}}^j(A, b, c, l, u, x) = \max\left\{V_{\text{solu}}(A, b, c, l, u, x), \max_{1 \leq j' < j} \left|x_{\sigma(j')} - \Phi_{j'}(A, b, c, l, u)\right|\right\},$$

for $(A, b, c, l, u) \in \iota^{-1}(\mathcal{D}_{\text{solu}} \backslash \mathcal{D}_{\text{foldable}}) \cap (\Phi_{\text{sort}} \circ \iota)^{-1}(\sigma)$ and $x \in \mathbb{R}^k \times \mathbb{Z}^{n-k}$. Similar to $V_{\text{solu}}$, $V_{\text{solu}}^j$ is also measurable respect to $(A, b, c, l, u)$ for any $x$ and is continuous with respect to $x$. For any $(A, b, c, l, u) \in \iota^{-1}(\mathcal{D}_{\text{solu}} \backslash \mathcal{D}_{\text{foldable}}) \cap (\Phi_{\text{sort}} \circ \iota)^{-1}(\sigma)$, the followings are equivalent:

- $\Phi_{\text{solu}}(\iota(A, b, c, l, u))_{\sigma(j)} < \phi$.

- $\inf\left\{x_{\sigma(j)} : x \in X_{\text{solu}}(\iota(A, b, c, l, u)), \, x_{\sigma(j')} = \Phi_{j'}(A, b, c, l, u), \, j' = 1, 2, \ldots, j - 1\right\} < \phi$.

- There exist $r \in \mathbb{N}_+$ and $x \in \mathbb{R}^k \times \mathbb{Z}^{n-k}$, such that $V_{\text{solu}}^j(A, b, c, l, u, x) = 0$ and that $x_{\sigma(j)} \leq \phi - 1/r$.

- There exists $r \in \mathbb{N}_+$, for any $r' \in \mathbb{N}_+$, $x_{\sigma(j)} \leq \phi - 1/r$ holds for some $x \in \mathbb{Q}^k \times \mathbb{Z}^{n-k}$ satisfying $V_{\text{solu}}^j(A, b, c, l, u, x) < 1/r'$.

Therefore, $\Phi_j^{-1}(-\infty, \phi)$ can be expressed in similar format as $\Phi_1^{-1}(-\infty, \phi)$, and is hence measurable. $\qquad\square$

## D  DETAILS OF THE NUMERICAL EXPERIMENTS AND EXTRA EXPERIMENTS

**MILP instance generation**  Each instance in $\mathcal{D}_1$ has 20 variables, 6 constraints and is generated with:

- For each variable, $c_j \sim \mathcal{N}(0, 0.01)$, $l_j, u_j \sim \mathcal{N}(0, 10)$. If $l_j > u_j$, then switch $l_j$ and $u_j$. The probability that $x_j$ is an integer variable is 0.5.
- For each constraint, $\circ_i \sim \mathcal{U}(\{\leq, =, \geq\})$ and $b_i \sim \mathcal{N}(0, 1)$.
- $A$ has 60 nonzero elements with each nonzero element distributing as $\mathcal{N}(0, 1)$.

Each instance in $\mathcal{D}_2$ has 20 variables, 6 equality constraints, and we construct the $(2k - 1)$-th and $2k$-th problems via following approach ($1 \leq k \leq 500$)

- Sample $J = \{j_1, j_2, \dots, j_6\}$ as a random subset of $\{1, 2, \dots, 20\}$ with 6 elements. For $j \in J$, $x_j \in \{0, 1\}$. For $j \notin J$, $x_j$ is a continuous variable with bounds $l_j, u_j \sim \mathcal{N}(0, 10)$. If $l_j > u_j$, then switch $l_j$ and $u_j$.
- $c_1 = \cdots = c_{20} = 0$.
- The constraints for the $(2k-1)$-th problem (feasible) is $x_{j_1} + x_{j_2} = 1$, $x_{j_2} + x_{j_3} = 1$, $x_{j_3} + x_{j_4} = 1$, $x_{j_4} + x_{j_5} = 1$, $x_{j_5} + x_{j_6} = 1$, $x_{j_6} + x_{j_1} = 1$.
- The constraints for the $2k$-th problem (infeasible) is $x_{j_1} + x_{j_2} = 1$, $x_{j_2} + x_{j_3} = 1$, $x_{j_3} + x_{j_1} = 1$, $x_{j_4} + x_{j_5} = 1$, $x_{j_5} + x_{j_6} = 1$, $x_{j_6} + x_{j_4} = 1$.

**MLP architectures** As we mentioned in the main text, all the learnable functions in GNN are taken as MLPs. All the learnable functions $f_{\text{in}}^V, f_{\text{in}}^W, f_{\text{out}}, f_{\text{out}}^W, \{f_l^V, f_l^W, g_l^V, g_l^W\}_{l=0}^L$ are parameterized with multilayer perceptrons (MLPs) and have two hidden layers. The embedding size $d_0, \cdots, d_L$ are uniformly taken as $d$ that is chosen from $\{2, 4, 8, 16, 32, 64, 128, 256, 512, 1024, 2048\}$. All the activation functions are ReLU.

**Training settings** We use Adam (Kingma & Ba, 2014) as our training optimizer with learning rate of 0.0001. The loss function is taken as mean squared error. All the experiments are conducted on a Linux server with an Intel Xeon Platinum 8163 GPU and eight NVIDIA Tesla V100 GPUs.

**Extra experiments on generalization** We conduct some numerical experiments on generalization for the random-feature approach. The GNN/MLP architecture is the same as what we describe before with the embedding size being $d = 8$. The size of the training set is chosen from $\{10, 100, 1000\}$ and the size of the testing set is 1000, where all instances are generated in the same way as $\mathcal{D}_2$, with the only difference that we set $c_1 = \cdots = c_{20} = 0.01$ instead of $c_1 = \cdots = c_{20} = 0$. The reason is that, $c_1 = \cdots = c_{20} = 0$ makes every feasible solution as optimal and thus the labels depend on the solver's choice. By setting $c_1 = \cdots = c_{20} = 0.01$, one can make the label depend only on the MILP problem itself, which fits our purpose, i.e., representing characteristics of MILP problems, better. In addition, our datasets for feasibility consists of 50% feasible instances and 50% infeasible ones, while other datasets are obtained by removing infeasible samples.

We report the error on training set and the testing set in Table 1. The generalization performance is good in our setting when the training size is relatively large. Although our experiments are still small-scale, these results indicate the potential of the random-feature approach in solving MILP problems in practice.

| Number of Training Samples | 10 | 100 | 1000 |
|---|---|---|---|
| Feasibility | | | |
| Training Error | 0 | 0 | 0.02 |
| Testing Error | 0.289 | 0.104 | 0.022 |
| Objective | | | |
| Training Error | 1.2e-15 | 1.1e-06 | 1.1e-06 |
| Testing Error | 5.9e-03 | 5.4e-06 | 1.1e-06 |
| Solution | | | |
| Training Error | 1.7e-03 | 7.2e-02 | 7.2e-02 |
| Testing Error | 2.1e-01 | 7.5e-02 | 7.3e-02 |

Table 1: Generalization for feasibility

