# OpenReview forum: "On Representing Mixed-Integer Linear Programs by Graph Neural Networks"
_ICLR.cc/2023/Conference — ICLR 2023 poster_

### Official Review · Reviewer_LV4F · 2022-10-21

**Confidence:** 4
**Correctness:** 4
**Technical Novelty And Significance:** 3
**Empirical Novelty And Significance:** 2
**Recommendation:** 6

**Clarity, Quality, Novelty And Reproducibility:**

The paper is clearly written. Mathematical notations are consistent throughout the paper, and follow well-established conventions, which make it easy to follow the discussion without much effort.

The identification of the negative example which current graph representation fails is clever and insightful. Given GNNs have been quite popular for MILP applications, I am surprised such a negative result did not already exist. Hence I highly value the novelty of this work. Authors also built upon standard tools for understanding GNN representative power, which makes the method principled. Beyond the identification of the negative example & characterization with foldability, however, the rest of the analysis seems standard.

The paper is mostly theoretic, so reproducibility is not a big concern. I wasn't able to find code, but replicating results is likely low effort given the simplicity of experiments.



**Strength And Weaknesses:**

While GNNs have become a popular method of encoding MILP problems, there has been lack of clarity on whether these GNNs are actually representative enough for their successful applications to downstream tasks such as neural diving or branching. This paper provides a clear example which this method might fail, and provides an insightful characterization. I believe such a finding will help practitioners understand limitations of current approaches in applying GNNs to MILPs.

Authors also build upon well-established approaches from the literature: The suggested remedy of adding random features is simple, but well supported by recent literature on GNNs.

The contribution of the paper is mostly theoretical rather than empirical, and unfortunately the paper is not very clear about the limitation of the analysis. Most importantly, authors fail to mention that the particular bipartite graph representation they introduce is a simplification of what was actually introduced in Gasse et al. (2019). The paper almost reads like this representation is exactly what Gasse et al and follow-up papers like Nair et al are using, whereas in reality these papers have a number of engineered features and therefore the suggested "unfoldability" is much less likely to happen. I still find authors' analysis to be insightful- good analysis requires simplification. However, authors should've clarified they are making simplifications.

I am also not sure adding random features is the right remedy to the problem. For sure, making every node color unique will allow WL test to always separate two graph instances. But on the other hand, when there are two equivalent MILP problems, WL test will now fail to recognize they are equivalent? Hence we seem to be losing statistical power from this "remedy", but this loss of statistical power is not characterized, not giving readers the sense of trade-off being made.

Authors could've provided much more insight on the practical usefulness of foldability if they used MILP problems from existing benchmarks. How many of MILP problems in the literature are actually foldable; is there any? If there are, how does that impact the performance of GNN-based feasibility classifiers and/or neural diving/branching models?

**Summary Of The Paper:**

Authors identify cases which a particular method of graph representations of mixed integer linear programming (MILP) problems shall not distinguish feasible problems from infeasible problems using Weisfeiler-Lehman (WL) test. Authors characterize a subset of indistinguishable MILP problems as "foldable", which means there are nodes with the same color from WL test. As a treatment, authors suggest to add a random features to make all nodes distinguishable. Synthetic data experiments demonstrate that random features do provide representation power to separate feasible MILPs from infeasible ones.

**Summary Of The Review:**

Being clear about limitations of the analysis, and bridging the gap between the analysis & actual practice will strengthen the paper.

---

> ### Author Response · Authors · 2022-11-15
> **Reply to Reviewer LV4F (Part 1/2)**
>
> Thank you very much for your helpful comments! Please find our responses below:
>
> **(Num. of features).** We agree that the bipartite graphs we use have fewer features than Gasse et al. (2019). Thank you for pointing this out, and we mention it clearly in the revised version (Page 3, footnote 3) as follows:
>
> > In (Gasse et al., 2019) and other related empirical studies, the feature spaces $\mathcal{H}^V,\mathcal{H}^W$ are more complicated than those defined in this paper. We only keep the most basic features here for simplicity of analysis.
>
> Also, we agree with you that foldable MILP-graphs may become unfoldable after adding more features, and we clearly point this out in the updated paper (the last paragraph of Section 4 at Page 7) as follows:
>
> > Let us also mention that the foldability of an MILP instance may depend on the feature spaces $\mathcal{H}^V,\mathcal{H}^W$. If more features are appended (see (Gasse et al., 2019)), i.e., $h_i^V$ and $h_j^W$ have more entries, and then a foldable MILP problem may become unfoldable. In addition, all the analysis here works for any topological linear spaces $\mathcal{H}^V,\mathcal{H}^W$, as long as $\Phi_{\text{feas}}$, $\Phi_{\text{obj}}$, and $\Phi_{\text{solu}}$ can be verified as measurable and permutation-invariant/equivariant mappings.
>
> **(Random features).** Yes, two isomorphic MILP-graphs may become non-isomorphic if random features are added. However, it does not conflict with the conclusion that GNNs can approximate $\Phi_{\text{feas}}, \Phi_{\text{obj}}, \Phi_{\text{solu}}$.
>
> Denote the two MILP-graphs in Figure 2 as $(G,H)$ and $(\hat{G},\hat{H})$. One could check that the WL test can surely distinguish these two instances with random feature: $\text{WL}_\text{MILP}(G,H,\omega) \neq \text{WL}_\text{MILP}(\hat{G},\hat{H},\omega)$ with probability one if $\omega$ is uniformly sampled from $\Omega$. On the other hand, **the WL test will almost surely separate two copies of the same MILP-graph with different random features**: $\text{WL}_\text{MILP}(G,H,\omega) \neq \text{WL}_\text{MILP}(G,H,\hat{\omega})$. However, **unlike the WL test, a GNN could be non-injective,** and one can train a GNN $F_R$ such that
>
> $$
> F_R(G,H,\hat{\omega}) \approx F_R(G,H,\omega) \neq F_R(\hat{G},\hat{H},\omega)\text{ with high probability}.
> $$
>
> In practice, one may sample several independent random vectors for each MILP instance, but this approach may raise training difficulty because it enlarges the number of training samples. A more simple practice is to generate one random feature vector for all MILP instances (both training and testing instances). This setting leads to efficiency in training GNN models, but the trained GNNs cannot be applied to MILP problems of different sizes. Such a trade-off also occurs in other GNN tasks (Loukas, 2019; Balcilar et al., 2021). In this paper, we adopt the latter to validate our theorems. How to balance the trade-off in practice will be an interesting future topic.
>
> **We will definitely add the above discussions in our next version of the manuscript.** We appreciate it that you and another reviewer wHAc point this out to improve the readability of this paper.

---

> ### Author Response · Authors · 2022-11-24
> **## Reply to Reviewer LV4F (Part 2/2)**
>
> **(Foldability).** Foldability is actually an important and common issue in MILP problems. To support this point, we conduct the WL test on the dataset named MIPLIB2017 [1] (https://miplib.zib.de/), which consists of $1064$ instances. Around 1/4 of the problems are "extremely" foldable, which means that the number of vertex colors is less than **half** of the number of vertices:
> * Num. of extremely foldable instances: 266/1064
>
> On foldable problems, GNNs may struggle to predict a meaningful neural diving or branching strategy. Let's consider the two instances in Figure 2. Since the WL test labels all the variables with a uniform color, the WL test or GNNs cannot distinguish the 6 variables. In neural diving, one usually trains a GNN to predict an estimated solution, while the estimated solution for the MILPs in Figure 2 will be eighter all ones or all zeros. In neural branching, one usually trains a GNN to predict a ranking score for each variable and use that score to decide which variable is selected to branch first. However, the outputs of GNNs on the MILPs in Figure 2 are tied, and one cannot get a meaningful ranking score. We would like to add the above discussions to the revised paper.
>
> A footnote: in the above paragraph, we abbreviate "GNNs defined in Section 2" to "GNNs." A practical GNN may involve more features as you pointed out.
>
> **(Summary).** As you suggested, "Being clear about limitations of the analysis, and bridging the gap between the analysis & actual practice will strengthen the paper." In our revised manuscript, we discussed the trade-off of the random-feature technique, the influence of foldability in practical settings, and the relationship between our theoretical results with the recent empirical studies in the literature. The three points can be found in Sections 5, Section 3, and Section 1, respectively. Thanks again for your valuable comments!
>
> [1] Gleixner, Ambros, et al. "MIPLIB 2017: data-driven compilation of the 6th mixed-integer programming library." Mathematical Programming Computation 13.3 (2021): 443-490.

---

### Official Review · Reviewer_mtYU · 2022-10-21

**Confidence:** 5
**Clarity, Quality, Novelty And Reproducibility:** This is a well-organized paper and ea…
**Correctness:** 4
**Technical Novelty And Significance:** 4
**Empirical Novelty And Significance:** 4
**Recommendation:** 8

**Strength And Weaknesses:**

Strong points:
1. Studying the representability of GNN for MILPs is a fundamental problem in this area， and this topic fits well with the conference's scope.
2. The construction of the counterexample is neat, and this example clearly shows the limitation of the representability of GNN for MILP problems.
3. Proving GNN has strong enough representation power by appending random features provides some theoretical explanations for this widely known trick.


Weak points:
1. Since the definition of foldable MILP depends on Algorithm 1, which is used to measure the separation power of GNN, the resulting theorems (Thm 4.2 ~ Thm 4.4) seem to be straightforward.
2. For the experiments, reporting training loss is not very meaningful from a practitioner's point of view. As you claim that adding random features is a good practical solution, can you also report the corresponding test loss?
3. Although the three questions raised in the paper are important, they are not quite practical. It will be better if the authors can link the results of this paper to answer whether GNN is able to predict the branching strategy which has been empirically studied recently.

**Summary Of The Paper:**

In this paper, the authors try to answer an interesting yet challenging question：
Is GNN a good representation model for MILP and whether it can predict important properties, eg., feasibility, for MILP problems?
They construct a simple example showing that there exist two MILPs such that one of them is feasible while the other one is not, but no GNN can distinguish them.
This example illustrates the limitation of the representability of GNN for MILP problems.
They further divide MILPs into two classes: foldable and unfoldable and show that GNN has strong enough representation power on unfoldable MILPS.
Finally, they show that by appending random features, GNN can also have strong enough representation power on foldable MILPS.


**Summary Of The Review:**

Overall, this is a good paper that tries to answer important questions on the topic of "learning to optimize".

---

> ### Author Response · Authors · 2022-11-15
> **Reply to Reviewer mtYU**
>
> Thank you very much for your encouraging comments! Please see our response to the weak points you mention below:
>
> **(Foldability).** In our settings, foldability is actually a concept only depending on the MILP instance. In the main text, we utilize the tool of the WL test to define foldability here for readability. In the revised version, we present a more complex but equivalent definition merely with the language of MILP in Definition A.1.
>
> **(Generalization).** Yes, we present some generalization results here, and we would like to add them to the appendix of this paper. To test the generalization of our trained models, we generate a testing set with $1000$ instances and three training sets with $10,100,1000$ instances, respectively. The testing instances are independent of those training instances. All the MILP instances here are **foldable**. The embedding size of the GNNs is $d=8$, and we append random features to the vertex features. We report the training and testing results in the following table. It's clear that **as long as the number of training samples is large enough, the generalization performance is good.**
>
> Feasibility:
> In both training and testing sets, there are $50\\%$ feasible instances and $50\\%$ infeasible ones.
> | Num. Training Samps. | 10    | 100   | 1000  |
> |----------------------|-------|-------|-------|
> | Training Error       | 0     | 0     | 0.02 |
> | Testing Error        | 0.289 | 0.104 | 0.022 |
>
> Objective:
> Only consider feasible instances and remove those generated infeasible ones.
> | Num. Training Samps. | 10       | 100      | 1000     |
> |----------------------|----------|----------|----------|
> | Training Error       | 1.2e-15 | 1.1e-06 | 1.1e-06 |
> | Testing Error        | 5.9e-03 | 5.4e-06 | 1.1e-06 |
>
> Solution:
> Only consider feasible instances and remove those generated infeasible ones.
> | Num. Training Samps. | 10       | 100      | 1000 |
> |----------------------|----------|----------|------|
> | Training Error       | 1.7e-03 | 7.2e-02 | 7.2e-02  |
> | Testing Error        | 2.1e-01 | 7.5e-02 | 7.3e-02  |
>
> Some details in the experiment. The approach to generating foldable MILP instances here is almost the same as that in our paper. The only difference is that the MILP instances here are generated with $c_1=c_2=\cdots=c_{20}=0.01$ instead of $c_1=c_2=\cdots=c_{20}=0$ in the paper. We would like to clarify the purpose here.
> * With $c_1=c_2=\cdots=c_{20}=0$ in a MILP, any feasible solutions are optimal solutions; hence the multiplicity of the solution set is great. We adopt this setting in the experiments of our paper and report training results in Figures 3 and 4. To make labels for GNNs, we call a MILP solver to obtain one of the optimal solutions. In this case, **the solution of a MILP not only depends on the MILP itself but also depends on the solver we use.** Although this setting is extremely hard from this expressiveness perspective, our results show that there exists a GNN that can approximate the solution to a given precision, which directly validates our theorems.
> * With $c_1=c_2=\cdots=c_{20}=0.01$, the generated MILP may still have multiple optimal solutions, but the multiplicity of the solution set is significantly reduced. In this case, the results in this rebuttal show that GNNs with random features not only have good expressive power but also be able to generalize well to instances that are similar to training samples.
>
> We added the above discussions in our revised paper.
>
> **(References).** Yes, on the second page of our revised paper, we  discuss more on the relationship between our theoretical results and those promising empirical studies:
> * The answers to (Q1), (Q2), and (Q3) serve as foundations of a more practical question: whether GNNs are able to predict branching strategies or primal heuristics for MILP?
> * Although the answers to (Q1) and (Q2) do not directly answer that question, they illustrate the possibility that GNNs can **capture some key information** of a MILP instance and **have the capacity** to suggest an adaptive branching strategy for each instance. To obtain a GNN-based strategy, practitioners should consider more factors: feature spaces, action spaces, training methods, generalization performance, etc., and some recent empirical studies show encouraging results on learning such a strategy. [Gasse et al, (2019); Gupta et al. (2020); Nair et al. (2020); Shen et al. (2021); Gupta et al. (2022);Qu et al. (2022); Scavuzzo et al. (2022); Khalil et al. (2022)]
> * The answer to (Q3) directly shows the possibility of learning primal heuristics. With proper model design and training methods, one could obtain competitive GNN-based primal heuristics. [Nair et al. (2020); Ding et al. (2020)]
>
> The above references have already been cited in our paper. We would highly appreciate it if you could point out references that are missed in our paper (if any).

---

### Official Review · Reviewer_ekKz · 2022-10-24

**Confidence:** 2
**Correctness:** 3
**Technical Novelty And Significance:** 2
**Empirical Novelty And Significance:** Not applicable
**Recommendation:** 1

**Clarity, Quality, Novelty And Reproducibility:**

Clarity:
	Most of the paper is clearly written but in some descriptions, contribution and reusing of others work is not clear. eg. Do the theorems of section 4 and 5 are contributions or reapplications?
Quality and Novelty:
	This paper is mostly about Graph theory and MILP and these are not in my area of expertise. Therefore, I am not able to judge the quality or the novelty of this paper. In my opinion this paper is not suitable for a machine learning conference. This paper should be submitted to a applied mathematics or graph conference.
Reproducibility: Seems reproducible as theorems proofs and details are provided in the appendix


**Strength And Weaknesses:**

Strength:
	1. The proposed technique of addition of random features
	2. Mathematics and details of the MILP representation into GNN
	3. Reapplication of theorem 3.1. for MILP - graphs
	4. Theorems of section 4 & 5
	5. Numerical experiments to validate proposed theories
Weaknesses:
	1. Typos e.g. "Fugure 2" on page 6
	2. It is not clear what is the contribution to the mathematics and theory and what is reapplication of existing work.
	3. Mainly the paper is about graph theory
	4. Although GNN is used in machine learning, the paper does seems to be doing machine learning
	5. No real world MILP scenarios are implemented or solved


**Summary Of The Paper:**

The paper presents a Graphical Neural Network (GNN) based method for the solution of Mixed-integer linear programming (MILP) optimization problem. The proposed method adds random features to the MILP based GNNs and prove is provided that this will provide a optimal solution for MILP. Details, from a reference,  are provided for the representation of an MILP into a weighted graph. GNN is built to represent the properties of the whole graph. Feasibility, Optimal objective value and Optimal solution mappings are provided along with the invariance and equivariance properties. A lot of theoretical description of mathematics is provided for the GNN based MILP solution. Feasibility through numerical experiments is  described.


**Summary Of The Review:**

	1. The paper seems sound mathematically and theoretically.
	2. Mostly the paper is about MILP and Graphs theory
	3. Other than GNN, no machine learning
	4. More suitable for graph or applied mathematics conference
	5. Not suitable for a machine learning conference

---

> ### Author Response · Authors · 2022-11-15
> **Reply to Reviewer ekKz**
>
> Thanks a lot for reading our paper and submitting your comments. We believe this paper is a **precise fit** for ICLR because it is about representation learning of a very difficult task --- solving MILPs --- using GNNs.
>
> Let us mention some previous ICLR papers that also study the representation power of GNNs:
> * Keyulu Xu, Weihua Hu, Jure Leskovec, and Stefanie Jegelka. How powerful are graph neural networks? In International Conference on Learning Representations, 2019.
> * Waiss Azizian and Marc Lelarge. Expressive power of invariant and equivariant graph neural networks. In International Conference on Learning Representations, 2021.
> * Floris Geerts and Juan L Reutter. Expressiveness and approximation properties of graph neural networks. In International Conference on Learning Representations, 2022.

---

### Official Review · Reviewer_wHAc · 2022-10-25

**Confidence:** 5
**Clarity, Quality, Novelty And Reproducibility:** See my comments below.
**Correctness:** 4
**Technical Novelty And Significance:** 4
**Empirical Novelty And Significance:** 3
**Recommendation:** 6

**Strength And Weaknesses:**

See my comments below.

**Summary Of The Paper:**

This paper study the representation power of GNN for mixed integer linear programming (MILP) problems.

**Summary Of The Review:**

This paper studies the representation power of GNN for mixed integer linear programming (MILP) problems.  The problem is important and fundamental. The analysis justifies the advantage of the random feature technique when solving MILP problems. The overall presentation is quite clear.  The proof combines the Stone-Weierstrass theorem (including its generalized version in Azizian and Lelarge’21) with domain knowledge of MILP.  The anaylsis is mostly sound and solid. However, I have the following questions.

**Major questions:** several important proof steps are shown in a hand-waving style without rigorous derivation. For instance:

1. Lemma A.8 is a very strong claim. I think it only holds for some specific graphs like bipartite graphs. Please write down the proof rigorously. The same suggestion also applies to the proof of Thm 5.1, page 16, the claim in the sentence "Therefore, similar to Lemma A.8, one can see for any....". Please write down the proof rigorously.

2. For the proof of Thm 5.1, page 16,  please verify that for any function f in the function class of  GNN, f(G1) = f(G2) if G1 and G2 (which are both equibed with random features) are isomorphic.  Please write down the proof rigorously (if this claim is correct.)

3.  Why is phi_feas continuous?

4. phi_solu( ) is an one-to-many mapping. It is not a function. I don't understand how to approximate this mapping using Lusin' & Stone-Weisterass Theorem, which only apply to measureable & continuous functions. Please explain more.

A follow-up question related to the above comments 1 and 2:

Regarding the proof of Thm 5.1, page 16, the authors apply Thm A.7 after verifying  "phi_feas(G, H ) =  phi_feas(Ghat, Hhat), for all (G, H,w )~ (Ghat,Hhat,what)". However, I dont think we can apply  Thm A.7 like this because (G,H) and (G, H, w) lie in different spaces.  From my understanding, by applying Thm A.7 correctly, we can only get "random-featured GNN" can approximate "phi_feas defined on random-featured  graphs".  But it seems unclear what is the relation between "phi_feas defined on random-featured  graphs" and "phi_feas defined on original  graphs". Please verify.

To briefly summarize, the proof of Thm 5.1, page 16 is quite dense and hand-waving. **The rigorous proof is missing and I doubt the correctness of the results about random features.  I would like to raise the score if authors address these concerns.**



**Other comments:**
1. The paper steps forwards an important first step to establish the feasibility of learning MILP by GNNs. However, the problem is far from being concluded. For MILP problems, another equally important (if not more important) application of GNN is to learn the algorithmic configuration such as the branching strategy, which is not covered in the script. Please add more relevant discussions.

2. Correct me if wrong: from my understanding,  the original WL test paint all the nodes in a single graph with the same initial color.  Why do we allow different initial colors (across nodes) in the script?

2. typo: page 6 lemma 3-2: Consider two MILP provlems ==> problems.

3. I would suggest the authors double-check all the \citep and \citet properly.

4. When encoding the MILP into a graph, how do you implement "<=, >=" and "\infty" in the node feature of the graph?


+++++++++++++++++++++**POST REBUTTAL** ++++++++++++++++++++++

Many thanks for the detailed reply from the authors. Most of my concern is addressed.  I am also convinced that the script has many differences with  the LP paper  so the ethic concern is also addressed. I briefly summarize the contribution of the script as follows:

1. Negative results on GNN for representing MILP
2. A candidate solution called “random-featured GNN“ is provided.

For me, 1 is not surprising because it is well known that GNN has limited distinguishing power, so it is possible that GNN might fail to capture certain info on MILP graph. However, it is good to see a counter-example to explicitly point out the limitation of GNN for representing MILP.  In summary, although the result is not surprising, I think this part is still worth sharing with the community.

As for 2, I think the contribution is limited due to the following reasons.

[R1] The analysis requires infinite-width random-featured GNN.  It seems unclear what is the advantage of random-featured GNN over MLP. Since infinite width MLP has universal approximation power, MLP can solve isomorphism problems and also it can represent MILP as well. Then the random-feature strategy seems not that important since you also require infinite width anyways.

[R2] The practical impact is unclear due to the toy experiments. Further, random-feature technique is not invented by the script, either.


By balancing the pros and cons, I raise my score to 6.

---

> ### Comment · Reviewer_wHAc · 2022-10-26
> **Some ethics concerns**
>
> The math technique in the current script is very similar to [1], which is a concurrent work focusing on LP problems instead of MILP.  I noticed [1] because the authors repeatedly cite many results and discussions in [1]. I check both papers carefully and I found that the crucial proof technique in the submission is very similar to [1]: With the help of theorems in [1], the current script is mostly about checking the analytical conditions to apply theorems in [1]. This process is still technical, but less significant. I would believe that both works are done by the same research group so it causes great trouble for my review process. For me,  if the proof is correct and sound, it is okay to accept either one of the LP or MILP paper. However, I do not think it is appropriate to accept both the LP paper and the MILP paper simultaneously due to the great similarity of these two works (but I am not sure if the LP paper is also submitted to ICLR 2023).  Anyhow, I would leave my concern here and leave the decision to AC.
>
> [1] Ziang Chen, Jialin Liu, Xinshang Wang, Jianfeng Lu, and Wotao Yin. On representing linear programs by graph neural networks. arXiv preprint arXiv:2209.12288, 2022.

---

> > ### Author Response · Authors · 2022-11-15
> > **Reply to the ethic concerns of Reviewer wHAc**
> >
> > Even if we used some results from [1], we believe that the difference is significant. It is proved in [1] that the limitation of GNNs does not effect representing LP problems, but our result is that MILPs do suffer from such limitation, and we propose some methods to resolve this issue. Thus, the stories of our manuscript and [1] are **entirely different**, though both works consider representation power of GNNs. In addition, some of our analysis is **highly nontrivial**. For example, the construction and properties of $\Phi_{\text{solu}}$ (see Appendix C).

---

> ### Author Response · Authors · 2022-11-15
> **Reply to Reviewer wHAc**
>
> Thank you very much for your helpful comments. Below are our reponses to your major questions:
>
> * Lemma A.9 (Lemma A.8 in the original version) actually holds for general graphs (not limited to bipartite graphs). Consider two unfoldable graphs $G$ and $\hat{G}$. Denote $\{v_1,v_2,\dots,v_n\}$ and $\{\hat{v}_1,\hat{v}_2,\dots,\hat{v}_n\}$ as the sets of vertices of $G$ and $\hat{G}$. Let $C(v_i)$ and $C(\hat{v}_i)$ be the final colors of $v_i$ and $\hat{v}_i$ generated by WL test. The definition of the unfoldability guarantees that $C(v_i)\neq C(v_j)$ and $C(\hat{v}_i)\neq C(\hat{v}_j)$ for all $i\leq j$. Suppose that $G$ and $\hat{G}$ cannot be distinguished by WL test, after applying some permutation, it holds that for any $i=1,2,\dots,n$ that $C(v_i) = C(\hat{v}_i)$, which implies that $v_i$ and $\hat{v}_i$ share the same neighbourhood information. Particularly, we have $E(v_i,v_j) = E(\hat{v}_i,\hat{v}_j)$ where $E(\cdot,\cdot)$ represent the edge. $G$ and $\hat{G}$ are thus isomorphic. We acknowledge that the proof is nontrivial and we elaborated it in the revised version. Meanwhile, we re-defined all the required preliminaries before proving Theorem 5.1, which allows us to directly apply Lemma A.9 in the proof of Theorem 5.1.
> * This is a good point. The reason is that a GNN is always permutation invariant/equivariant by the definition, no matter how we choose the spaces for vertex features, i.e., $\mathcal{H}^V$ and $\mathcal{H}^W$. If we add a random feature, we are basically using new features spaces, i.e., replacing $\mathcal{H}^V$ and $\mathcal{H}^W$ by $\mathcal{H}^V\cup[0,1]$ and $\mathcal{H}^W\cup[0,1]$. A GNN defined on these new spaces are still permutation invariant/equivariant. We clarified this in the revised version.
> * $\Phi_{\text{feas}}$ is not continuous in general. We prove in Lemma A.4 that it is measurable and use that lemma to prove Theorems A.2. In the proof of Theorem 5.1, $\Phi_{\text{feas}}$ restricted on $\mathcal{D}\times \Omega_\epsilon$ is continuous since $\mathcal{D}$ is a discrete finite set and $\Phi_{\text{feas}}(G,H,\omega)$ is independent of $\omega\in\Omega_\epsilon$ (the feasibility does not depend on the random features). Note that any mapping defined on a discrete finite domain is continuous. We rephrased the proof of Theorem 5.1 to clarify this point.
> * Although a MILP instance may have multiple solutions, the optimal solution mapping $\Phi_{\text{solu}}$ we construct returns **only one** solution among all optimal solutions. We have included the detailed construction and definition in Appendix C. We have also proved in Appendix C that $\Phi_{\text{solu}}$ is measurable and equivariant. We would like to briefly summarize our proof line here: We first sort all the vertices in an MILP-graph with a lexicographical order, and then, among all the optimal solutions, we pick the first one. Since we assume that the colors of any two vertices are distinct, it holds that the solution picked in this approach is unique.
>
> Our response to your follow-up question:
> * $\Phi_{\text{feas}}$ with random features can be defined by $\Phi_{\text{feas}}(G,H,\omega):=\Phi_{\text{feas}}(G,H)$ for all $\omega \in [0,1]^{m+n}$. The reason is that the feasibility of the MILP problem does not depend on the the random features added. This is also the reason why verifying $\Phi_{\text{feas}}(G,H) = \Phi_{\text{feas}}(\hat{G},\hat{H})$ is equivalent to verifying $\Phi_{\text{feas}}(G,H,\omega) = \Phi_{\text{feas}}(\hat{G},\hat{H},\hat{\omega})$. We have revised the proof of Theorem 5.1 and added some preliminaries before the proof.
>
> Our responses to your other comments:
> * Yes, you are correct. We will add some discussions on the relationship between our theoretical results and those recent empirical studies on learning branching strategies, learning primal heuristics, etc.
> * Yes, the original WL test use the same initial color for all vertices, because it is for graphs without vertex features. However, in those graphs with vertex features, one usually assign different initial colors for vertices with different features. Thus, we initialize the colors according to the features. (refer to equation (1.3) in [1])
> * Thanks! We fixed the typo.
> * Thanks! We will check the citation styles.
> * We implement $\{\leq, =,\geq\}$ by $\{-1,0,1\}$, and $\infty$ can be implemented by adding one feature in the bounds to indicate whether it is finite or not.
>
> We believe that our theoretical results are all correct, though we did not include some details in the proofs. Thank you again for pointing out these issues, which is definitely helpful us for improving the manuscript!
>
>
> [1] Jegelka, Stefanie. (2022) "Theory of Graph Neural Networks: Representation and Learning."

---

> > ### Comment · Reviewer_wHAc · 2022-11-21
> > **A follow-up question around Lemma A.9**
> >
> > Many thanks to the detailed reply and revision. I am sorry that I still don't quite understand the claim around Lemma A.9
> >
> > It seems that the following claims are both true:
> >
> > Claim 1: Lemma A.9 does not dependent on the structure of bipartite graph,
> >
> > Claim 2: random-feature strategy can turn any foldable graphs into unfoldable, doesn’t seem to dependent on the structure of bipartite graph, either.
> >
> > Then combining together,  does it mean random-featured GNN can solve graph isomorphism problem? Isn’t that too good to be true?  Is anything wrong with Claim 1 or 2?

---

> > > ### Author Response · Authors · 2022-11-21
> > > **Reply to Reviewer wHAc**
> > >
> > > Thanks a lot for your time and for your comments! Yes, both Claim 1 and Claim 2 are correct. However, they **cannot** solve the graph isomorphiam problem. Suppose that we are given two isomorphic graphs, say $(G,H)$ and $(\hat{G},\hat{H})$, and we randomly generate two sets of random features $\omega$ and $\hat{\omega}$. Then with probability one, there is no permutation that can match $\omega$ with $\hat{\omega}$. Therefore, $(G,H,\omega)$ and $(\hat{G},\hat{H},\hat{\omega})$ are **no longer isomorphic**, which means that we cannot solve the graph isomorphiam problem using random feature approach since it almost surely detects every pair of graphs as nonisomorphic. Plese let us know if this answers your question. Many thanks!

---

> > > > ### Comment · Reviewer_wHAc · 2022-11-21
> > > > **What if we apply one set of random feature on both graphs?**
> > > >
> > > > Thanks for the prompt reply. Correct me if wrong: according to the reply, it seems that using one set of random features on both graphs can solve the issue you brought up.
> > > >
> > > > Then my question is: does it mean that we can solve the graph isomorphism problem (for general graphs, not limited to bipartite graphs)  if we use one set of random features on both graphs?
> > > >
> > > > In other words, can we solve the graph isomorphism problem by simply modifying "WL test" to "WL test with one set of random features on both graphs"?

---

> > > > > ### Author Response · Authors · 2022-11-21
> > > > > **Using the same set of random features may still break the isomorphism**
> > > > >
> > > > > Thanks for your question! The graph isomorphism problem still **cannot** be solved by using the same random features for different graphs.
> > > > >
> > > > > Let us consider two isomorphic graphs $G$ and $\hat{G}$ that are general graphs with $n$ vertices. Then there exist permutations $\sigma\in S_n$ such that $\sigma\ast G = \hat{G}$. We assume that $\sigma\neq\text{id}$ since otherwise detecting the isomorphism is trivial. Now let both graphs be appended with the same random features $\omega$. If we want to maintain the property  $\sigma\ast (G,\omega) = (\hat{G},\omega)$, then the random features should satisfy $\sigma(\omega) = \omega$. However, if $\omega$ is generated randomly, then almost surely we will have $\sigma(\omega) \neq \omega$, i.e., $\sigma\ast (G,\omega) \neq (\hat{G},\omega)$. Therefore, $(G,\omega)$ and $(\hat{G},\omega)$ are almost surely **no longer isomorphic** unless $G=\hat{G}$, i.e., $\sigma = \text{id}$. So we cannot solve the graph isomorphism problem with this approach.

---

> > > > > > ### Comment · Reviewer_wHAc · 2022-11-22
> > > > > > **Thanks for the reply. I think more relevant discussion in the script is needed**
> > > > > >
> > > > > > Thanks for the reply. I think now I have  better understanding on the random feature strategy
> > > > > >
> > > > > > Fact 1:  the random-feature strategy will make WL test even worse than before.
> > > > > >
> > > > > > This fact seems like bad news for the random-feature strategy.  It is natural to make readers think that “random-feature strategy will make GNN worse on learning phi_feas”. Actually, I noticed another reviewer posted a similar question and I find it very reasonable (since the authors did not explain the relation and difference.)
> > > > > >
> > > > > >
> > > > > > However, I now believe that  Fact 1 actually has nothing to do with “whether random-feature GNN can  learn phi_feas  or not”
> > > > > > This is because (correct me if wrong): unlike WL,  GNN is not necessarily injective:  after applying random feature on two isomorphic MILP-graph G1 and G2,  GNN (G1) and GNN(G2) can still be the same even if G1 is no longer isomorphic G2. Therefore, GNN can still learn the feasibility mapping on G1 and G2.
> > > > > >
> > > > > >
> > > > > > Technically, WL test is just an intermedia tool to define the equivalence relation ~. We can dump it after the ~ is defined. In fact, for the random-feature part, the equivalence ~ becomes isomorphism (Lemma A.9) so the whole random-feature part actually has nothing to do with WL test at all!  In other words, **“the ability of random-featured GNN to learn phi_feas” has nothing to do with “the ability of random-featured WL to distinguish graphs”**. These two topics are actually completely independent. However, I would expect most readers to be confused about this point if the authors do not point it out explicitly.  I would suggest the authors add more discussions in the script.
> > > > > >
> > > > > >
> > > > > > Regarding the author's answer to reviewer LV4F “ In our experiments, we generate one random feature vector for all instances, and it does not suffer from the problem you described.”. I don't think this answer is correct, because  “one random feature for all” still suffers from the problem pointed out by LV4F“WL  mis-distinguish two isomorphic graphs”.  But as discussed above, it doesn’t matter how WL will behave.  I would suspect that the training difficulty is the actual reason of using one “one random feature for all”.

---

> > > > > > > ### Author Response · Authors · 2022-11-24
> > > > > > > **We will add some discussions in the updated version (Part 1/2)**
> > > > > > >
> > > > > > > Thank you very much for your comments and for your suggestions. We acknowledge that we should add some discussions to avoid misunderstanding and we will do this in the updated version.
> > > > > > >
> > > > > > > Yes, we agree with you that the random-feature approach will almost surely make WL test detect every pair of graphs as non-isomorphic. Thus, this approach is definitely not suitable for the graph isomorphism problem. In other words, **with random features, WL test becomes worse in detecting isomorphic graphs** (isomorphic graphs may be detected as non-isomorphic with random features), **but becomes better in separating different graphs** (non-isomorphic graphs will almost surely be detected as non-isomorphic). Also, you are correct that even if WL test becomes worse in the isomorphism test, random-featured GNNs can still approximate $\Phi_{\text{feas}}$. **We will rewrite the paragraph following Theorem 5.3 at Page 8 as follows:**
> > > > > > >
> > > > > > > > Some discussions. Denote the two MILP-graphs in Figure 2 as $(G,H)$ and $(\hat{G},\hat{H})$. One could check that the WL test can distinguish these two instances with random feature: $\text{WL}_\text{MILP}(G,H,\omega) \neq \text{WL}_\text{MILP}(\hat{G},\hat{H},\omega)$ with probability one if $\omega$ is uniformly sampled from $\Omega$. On the other hand, the WL test will almost surely separate two copies of the same MILP-graph with different random features: $\text{WL}_\text{MILP}(G,H,\omega) \neq \text{WL}_\text{MILP}(G,H,\hat{\omega})$. However, unlike the WL test, a GNN could be non-injective, and one can train a GNN $F_R$ such that
> > > > > > > $$
> > > > > > > F_R(G,H,\hat{\omega}) \approx F_R(G,H,\omega) \neq F_R(\hat{G},\hat{H},\omega)\text{ with high probability}.
> > > > > > > $$
> > > > > > > > In practice, one may sample several independent random vectors for each MILP instance, but this approach may raise training difficulty because it enlarges the number of training samples. A more simple practice is to generate one random feature vector for all MILP instances (both training and testing instances). This setting leads to efficiency in training GNN models, but the trained GNNs cannot be applied to MILP problems of different sizes. Such a trade-off also occurs in other GNN tasks (Loukas, 2019; Balcilar et al., 2021). In this paper, we adopt the latter to validate our theorems. How to balance the trade-off in practice will be an interesting future topic.
> > > > > > >
> > > > > > > Thanks again for pointing the above valuable comment! We would like to provide another perspective to understand rand-feature here, which is actually our previous answer to Reviewer LV4F. Since $\omega$ is random, $F_R(G,H,\omega)$ is a coresponding random variable. Although it holds almost surely that $F_R(G,H,\omega) \neq F_R(\hat{G},\hat{H},\hat{\omega})$ even if $(G,H)$ is isomorphic to $(\hat{G},\hat{H})$, one may check $F_R(G,H,\omega)$ and $F_R(\hat{G},\hat{H},\hat{\omega})$ **follow the same distribution** if $(G,H)$ is isomorphic to $(\hat{G},\hat{H})$.

---

> > > > > > > ### Author Response · Authors · 2022-11-24
> > > > > > > **We will add some discussions in the updated version (Part 2/2)**
> > > > > > >
> > > > > > > However, we do not think “the ability of random-featured GNN to learn phi_feas” has nothing to do with “the ability of random-featured WL to distinguish graphs”. Consider the following event:
> > > > > > >
> > > > > > > > Event (A): The random vector $\omega$ for each MILP-graph is zero: $\omega^V_i = \omega^W_j = 0, \forall i \in \{1,2,\cdots,m\}, j \in \{1,2,\cdots,n\}$.
> > > > > > >
> > > > > > > **When event (A) happens, the limit of the WL test will surely affect the approximation power of GNN. Although the probability of (A) is zero, we could not ignore (A) in a rigorous proof.** Due to this consideration, we define the set $\Omega_F$ in equation (B.3), where $\Omega_F$ includes all events that are similar to (A). Since $\Omega \backslash \Omega_F$ may not be compact, we select a compact set $\Omega_{\epsilon} \subset \Omega \backslash \Omega_F$ such that $P(\Omega_{\epsilon})>1-\epsilon$ and then we apply Theorem A.8. This is why we have an $\epsilon$ in the conclusions. **If $\Omega_F$ was ignored, one would ignore the $\epsilon$ in the conclusion, which is not correct.**
> > > > > > >
> > > > > > > Furthermore, we will add the following discussions in the appendix to avoid confusing readers who read our proofs:
> > > > > > >
> > > > > > > > Theorem A.8 actually depends on the separation power of the WL test. Assumption (A.1) means that, to apply Theorem A.8, one have to show that the WL test can separate any MILP-graph pairs $(G,H), (\hat{G},\hat{H})$ as long as they have distinct $\Phi$ values: $\Phi(G,H) \neq \Phi(\hat{G},\hat{H})$. This is why we define the WL test and study its separation power. With the additional random feature technique, the separation power of the WL test is strengthened, so that we can obtain stronger conclusions.
> > > > > > >
> > > > > > > Yes, we use the same random features for all instances in the experiments since it is easier to train. We rephrase our reply to Reviewer LV4F based on the above discussions.
> > > > > > >
> > > > > > > We hope the above discussions fix your concerns, and **we will definitely update our draft as the above discussions in the final version**. However, the deadline to modify the pdf file in the system is Nov. 18, and we are not able to revise the pdf now. If you think an updated pdf is necessary, we would like to provide an anonymous link to access our updated paper (under the permission of the area chair).

---

> > > > > > > > ### Comment · Reviewer_wHAc · 2022-11-24
> > > > > > > > **Many thanks to the detailed reply. I have raised my score**
> > > > > > > >
> > > > > > > > Thanks for the detailed reply. My questions about the random-feature part are now addressed. I have updated my summary in the main panel (titled "post-rebuttal") and I also raised my score to 6.

---

> > > > > > > > > ### Author Response · Authors · 2022-11-28
> > > > > > > > > **Many thanks**
> > > > > > > > >
> > > > > > > > > Thank you so much for your great efforts and detailed comments in the review, and many thanks for updating the score!

---

### Public Comment · ~Fanchen_Bu1 · 2022-11-07
**Two suspiciously similar submissions**

https://openreview.net/forum?id=4gc3MGZra1d
https://openreview.net/forum?id=cP2QVK-uygd

The above two submissions look suspiciously similar.
Would the authors like to explain the similarity and elaborate on the differences?

---

> ### Author Response · Authors · 2022-11-07
> **differences and similarities**
>
> Dear Fanchen Bu,
>
> 1. The two papers prove different results. One paper proves that sufficiently large GNNs can solve linear programs (LPs). The other paper shows that **no** GNNs can distinguish certain pairs of mixed-integer programs (MILPs) in which one is feasible and the other is infeasible. The latter paper also introduces remedies. They have different contributions.
>
> 2. The MILP paper quotes some results from the LP paper with proper crediting.
>
> 3. The two papers use similar notation and have similar writing styles, but there is no plagiarism.
>
> Please let us know if you want us to clarify anything. If you still have any suspicions, we would ask you to articulate them so we can precisely address them.
>
> Respectfully,
>
> Authors

---

> > ### Public Comment · ~Fanchen_Bu1 · 2022-11-07
> > **Thank you for your prompt reply**
> >
> > Dear authors,
> >
> > Thank you for your prompt reply!
> > My main concerns are that the two submissions share too much writing, e.g., equations, examples, figures, and algorithms.
> > As you have claimed, the two papers show different theoretical results and there is no plagiarism.
> > Since it is hard to delineate the border of plagiarism precisely and I have no right to do so,
> > I would like to trust your academic integrity and leave the judgment to the reviewers.
> >
> > Best,
> >
> > Fanchen

---

> > > ### Author Response · Authors · 2022-11-07
> > > **The similarities are with proper citation and crediting**
> > >
> > > Dear Fanchen,
> > >
> > > 1. We think that most similarities you find are in Section 2 Preliminaries. This section is for introducing definitions and notations, not for our results or contributions. We clearly write at the beginning of Section 2 that "**Our notations and definitions follow ...**" with proper citation and crediting.
> > >
> > > 2. Another similarity is the Algorithm 1 in Section 3. We also clearly write "**that follows the same lines as in ...**" with proper citation and crediting. You can find this sentence in the last paragraph of Page 5.
> > >
> > > Please let us know if we've addressed your concerns. Many thanks!
> > >
> > > Best,
> > >
> > > Authors

---

> > > > ### Public Comment · ~Fanchen_Bu1 · 2022-11-07
> > > > **Thank you for your reply again**
> > > >
> > > > Thanks!
> > > >
> > > > I have read the two submissions again and got a deeper understanding.
> > > > The differences, especially in the theoretical results, are significant.
> > > > And I fully acknowledge that the authors have cited another submission in the manuscript.
> > > > I just still think that verbatim references from another paper are not good practices in academia.
> > > > An example would be the E2V-SDE scandal that happened earlier this year.
> > > >
> > > > I believe that the authors would consider this and refrain from verbatim references as much as possible in the revised paper.
> > > > And I hope that I did not waste your precious time during the rebuttal period.
> > > >
> > > > Best,
> > > >
> > > > Fanchen

---

> > > > ### Author Response · Authors · 2022-11-16
> > > > **Reply to Fanchen**
> > > >
> > > > We definitely understand that when using the same sentences from another paper, one should cite the source and add quotation marks. However, when writing mathematical materials (equations, theorems, etc), quotation marks are not required, as long as the source is clearly cited; otherwise, the writing will be broken significantly. As an example, numerous papers on optimization define $\mu$-strong convexity and $L$-smoothness using almost the same sentences and we believe this is not verbatim plagiarism.
> > > >
> > > > Let us also mention that, there is a huge difference between **verbatim plagiarism** and **using state-of-the-art/standard/comfortable preliminaries**. The latter is just for the convenience of writers and readers. Note that preliminaries (like defining $\mu$-strong convexity and $L$-smoothness) are not contributions, ideas, or results of research.
> > > >
> > > > BTW, we found that some notations in Section 2.2 of our original submission are not needed in the proof. So we rewrite this part to make the notations simpler and more reader-friendly. Hope this resolves your concerns to some extent.
> > > >
> > > > Best,
> > > >
> > > > Authors

---

### Decision · Program_Chairs · 2023-01-20

**Decision:**

Accept: poster

**Justification For Why Not Higher Score:**

The major issue lies in the random-feature trick. I explain as follows.
1.In theory:  The analysis of the random feature technique requires infinite-width GNNs, which is  not practical. A better analysis should work on GNN with finite number of parameters.
2.In practice: The practical value of the random feature trick remains unclear. As mentioned in the script, the random feature technique requires that the testing MILP has the same structure as the training MILP. This condition rarely holds in real-world tasks. I think a better theoretical paper should provide an analysis of a more practical trick.

**Justification For Why Not Lower Score:**

The paper shows that vanilla GNNs are not powerful enough to represent MILP problems. They further prove that the random feature strategy can overcome this issue. Most reviewers agreed that this paper made a good contribution to the L2O community. Even if this work has some limitations, its contribution outweighs its disadvantages.

**Metareview: Summary, Strengths And Weaknesses:**

This paper studies the representation power of GNN on mixed integer linear programming (MILP) problems. The paper shows that vanilla GNNs are not powerful enough to represent MILP problems. They further prove that the random feature strategy can overcome this issue. Most reviewers agreed that this paper made a good contribution to the L2O community. The major issue lies in the random-feature trick. I explain as follows.
1.In theory:  The analysis of the random feature technique requires infinite-width GNNs, which is  not practical. A better analysis should work on GNN with finite number of parameters.
2.In practice: The practical value of the random feature trick remains unclear. As mentioned in the script, the random feature technique requires that the testing MILP has the same structure as the training MILP. This condition rarely holds in real-world tasks. I think a better theoretical paper should provide an analysis of a more practical trick.
Even if this work has some limitations, its contribution outweighs its disadvantages. For this reason, I recommend accepting this paper.


**Note From Pc:**

if the above contains the word "oral" or "spotlight" please see: "oral" presentation means -> notable-top-5% and "spotlight" means -> notable-top-25%. As stated in our emails, we are disassociating presentation type from AC recommendations